

# Simulated retreat of Jakobshavn Isbræ during the 21st century

Xiaoran Guo[1], Liyun Zhao[1], Rupert Gladstone[2], Sainan Sun[3], John C. Moore[1,2,4]

[1]College of Global Change and Earth System Science, Beijing Normal University, Beijing 100875, China

[2]Arctic Centre, University of Lapland, P.O. Box 122, 96101 Rovaniemi, Finland

[3]Laboratoire de Glaciologie, Université libre de Bruxelles, Brussels, Belgium

[4]CAS Center for Excellence in Tibetan Plateau Earth Sciences, Beijing 100101, China

Correspondence to: John C. Moore (john.moore.bnu@gmail.com)

# Abstract

The early in the 21st century retreat of Jakobshavn Isbræ, one of Greenland's largest outlet glaciers, into its over-deepened bedrock trough was accompanied by acceleration to unprecedented ice-stream speeds. Such dramatic changes suggested the possibility of substantial mass loss over the rest of this century. Using a three-dimensional ice-sheet model with parameterizations to represent the effects of ice mélange buttressing, crevasse-depth-based calving and submarine melting, we can reproduce its recent evolution. The model can accurately replicate its inter-annual variations in grounding line and terminus position, including new modes of seasonal fluctuations that emerged after arriving at the over-deepened basin and the disappearance of a persistent floating ice shelf. The shear margin induced decreases in ice viscosity we simulate are particularly important in reproducing the large observed inter-annual changes in terminus velocity. We use this model to project Jakobshavn's evolution over this century when forced by the IPCC RCP4.5 climate scenario and simulated by ocean temperatures from 7 Earth System Models along with surface runoff derived from RACMO. In our simulations, Jakobshavn's grounding line continues to retreat ~ 18.5 km by





the end of this century with total mass loss of ~ 2030 Gt (5.6 mm sea-level-rise equivalent). Despite
the relative success of the model in simulating the recent behavior of the glacier, the model does not
simulate winter calving events that have become relatively more important.

# 1 Introduction

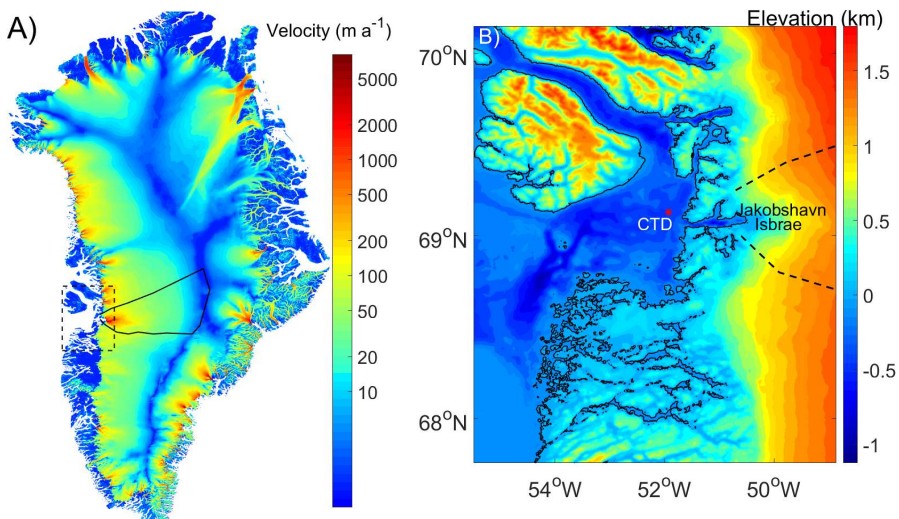

**Figure 1. A) Greenland ice sheet flow speeds from Joughin et al. (2018), with the Jakobshavn drainage basin outlined by the solid black line and the area shown in panel B by the dashed box. B) Ilulissat Fjord and Disko Bay bathymetry from Jakobsson et al. (2012), with the CTD (Conductivity Temperature Depth) site used for ocean temperature here marked by the red star.**

Jakobshavn Isbræ (Fig. 1) is Greenland's largest and fastest outlet glacier, with speeds of up to 17
km $a^{-1}$ (Joughin et al., 2014). Jakobshavn Isbræ drains ~ 6.5 % of the Greenland Ice sheet (Krabill
et al., 2000), and it alone contributed ~ 1 mm to global sea-level rise between 2000 and 2011 (Howat
et al., 2011). Since 1997, measurements indicate that the water entering Ilulissat Fjord where
Jakobshavn Isbræ terminates, is about 1.1 ºC warmer than it was during 1987-1991 (Holland et al.,
2008). This rise in water temperature coincided with the onset of dramatic thinning, speedup and



retreat of Jakobshavn Isbræ. By 2003 its velocity near the grounding line had reached ~ 12.6 km a⁻
¹, more than double that of 1992, and the floating ice mélange in the fjord had disintegrated (Joughin
et al., 2004). From 2005 to 2007, as it retreated inland, seasonal fluctuations in velocity 4 km inland
from the calving front amounted to $\pm$ 1 km a⁻¹. The winter slowdowns and summer accelerations
occurred in tandem with the calving front winter advance and summer retreat. By 2012 the seasonal
velocity fluctuations 4 km upstream from the calving front was nearly $\pm$ 8 km a⁻¹ and the grounding
line of Jakobshavn Isbræ had reached the bottom of a sub-glacial bedrock trough after years of
down-slope migration (Joughin et al., 2014).
Before 1997, Jakobshavn possessed a ~ 15 km long floating ice mélange in front of its terminus ice
cliff and experienced submarine melting on its ice-ocean interface (Amundson et al., 2010). After
1998 the terminus became more crevassed, coinciding with acceleration of the glacier, implying that
weakened buttressing had triggered its dramatic speed-up. A thinning rate of $230 \pm 50$ m a⁻¹ between
the summers of 1984 and 1985 was deduced from photogrammetric surveys, mostly due to
submarine melting (Motyka et al., 2011). The floating tongue thickened during the mid-1990s
followed by progressive thinning after 1997 (Motyka et al., 2011). From 1997 to 2008, the average
ocean temperature was 1.1°C higher than during the period 1980 – 1991, which raised its thinning
rate substantially, affecting the whole ice mélange, and the ice shelf eventually collapsed in 2003.
Many lines of evidence suggest that warm water was responsible for the submarine melting beneath
the ice mélange and ice-shelf, brought by a buoyancy-driven, overturning circulation in Ilulissat
fjord (Gladish et al., 2015).
Jakobshavn, in common with most outlet glaciers in Greenland, flows through a narrow, deeply
incised bedrock trough at a far faster rate than the ice surrounding it (Joughin et al., 2010). Gravity
surveys suggest a deep layer of soft till underlies much of the Jakobshavn trough (Block and Bell,



59 2011). This soft bed provides almost no resistance to ice flow and basal shear stress maps show that

60 most of the gravitational driving force on the glacier is balanced by lateral drag (Shapero et al.,

61 2016).

62 Basal drag decreased from 1995 to 2006 (Habermann et al., 2013), possibly due to fast thinning that

63 reduced the effective pressure, that is the ice overburden minus water pressure, at the bed. The

64 effective pressure distribution under the glacier is important to basal drag and must be zero at the

65 grounding line as it begins to float. Several sliding parameterizations (also termed sliding relations

66 or sliding laws) have been used in the literature that assume basal drag depends on sliding speed

67 (so-called Weertman sliding), or on effective pressure (Schoof, 2010; Gagliardini et al., 2014). Tsai

68 et al. (2015) introduced a combined Weertman and Coulomb sliding law based on effective pressures

69 with a boundary layer at the grounding line; this has a higher scaling of ice flux with grounding-line

70 thickness compared with the Weertman. However, in the Jakobshavn case, both Weertman and

71 Coulomb sliding produce very similar fluxes because the basal shear stresses along the main trough

72 are typically only 2 % of the driving force.

73 Simulations using a flow-band model with a crevasse-depth-based calving parameterization (Vieli

74 et al., 2011) demonstrated that loss of buttressing from the disintegration of its floating ice mélange

75 or enhanced submarine melting could have triggered the dramatic changes seen in Jakobshavn Isbræ

76 at the end of the 20th century. Later work (Muresan et al., 2016), using a simple calving model with

77 dependence on the strain field at the terminus was able to reproduce the inter-annual retreat of

78 Jakobshavn Isbræ until 2009, when the terminus arrived at the beginning of the reverse sloping bed.

79 But retreat after 2010 was not captured by their model, and neither was the seasonal fluctuation in

80 terminus position.

81 In this paper we use a three-dimensional ice-flow model with a treatment of calving that successfully



tracks the seasonal terminus position and its retreat into the over-deepened basin. We use historic
observations of ocean temperature as forcing and ice tongue thinning rate to scale submarine melting
rates for our model and thence make future projections. Our aim is to track the evolution of
Jakobshavn Isbræ through the 21st century under a specific climate forcing scenario. In Section 2
we describe the approach and calibration of our model, Section 3 shows the simulations for the
period to 2100 under the IPCC RCP4.5 scenario (Moss et al., 2010), Section 4 is a discussion of our
method with reference to other studies and suggestions for improvements, and we conclude in
Section 5.

# 2 Methods and data

## 2.1 Ice sheet model

We model Jakobshavn Isbræ using the BISICLES continuum ice sheet dynamics model that is based
on the vertically integrated stress balance formulation of Schoof and Hindmarsh (2010), which treats
longitudinal and lateral stresses as depth-independent, but allows for vertical shear in the nonlinear
rheology (Cornford et al., 2013). BISICLES is particularly useful for Jakobshavn Isbræ as it uses
block-structured finite volume discretization with adaptive mesh refinement (Cornford et al., 2013)
allowing for high resolution modeling of critical sections of the glacier. Jakobshavn Isbræ is fed by
a ~ 400 km long and extensive drainage basin (Fig. 1), but the fast flow area is only around 10 km
in width.
We assume Jakobshavn Isbræ to be in hydrostatic equilibrium, thus the upper surface elevation $s$ is
$$s = \max\left[h + b, \left(1 - \frac{\rho_i}{\rho_w}\right)h\right], \quad (1)$$





where $\rho_i$ and $\rho_w$ are the densities of ice and ocean water, $h$ is ice thickness and $b$ is bedrock
elevation relative to sea level. The ice thickness evolves in time as
$\frac{\partial h}{\partial t} + \nabla \cdot [\boldsymbol{u} h] = M_s - M_b,$ (2)
where $M_s$, $M_b$ are surface mass balance (SMB) and submarine melt rate respectively and $\boldsymbol{u}$ is the
depth-independent horizontal velocity. No basal melting over the grounded area is allowed. The
velocity $\boldsymbol{u}$ satisfies an approximate stress balance equation
$\nabla \cdot [\phi h \bar{\mu} (2\dot{\boldsymbol{\epsilon}} + 2\mathrm{tr}(\dot{\boldsymbol{\epsilon}})\mathbf{I})] - \boldsymbol{\tau}^b = \rho_i g h \nabla s,$ (3)
where $\mathbf{I}$ is the identity tensor, $s$ is the ice surface elevation, $g$ is the acceleration due to gravity, $\dot{\boldsymbol{\epsilon}}$ is
the horizontal strain-rate tensor defined by
$\dot{\boldsymbol{\epsilon}} = \frac{1}{2}[\nabla \boldsymbol{u} + (\nabla \boldsymbol{u})^{\mathrm{T}}],$ (4)
and $\boldsymbol{\tau}^b$ is the basal shear stress. The vertically integrated effective viscosity $\phi h \bar{\mu}$ is given by
$\phi h \bar{\mu}(x, y) = \phi \int_{s-h}^{s} \mu(x, y, z)\mathrm{d}z,$ (5)
where the vertically varying effective viscosity $\mu$ includes a contribution from vertical shear and
satisfies
$2\mu A(T)(4\mu^2 \dot{\boldsymbol{\epsilon}}^2 + |\rho_i g(s - z)\nabla s|^2)^{(n-1)/2} = 1,$ (6)
where $n$ is the flow rate exponent, set to 3 in the current study, and $A(T)$ is the rate factor, dependent
on the ice temperature $T$ through an Arrhenius law (Cuffey and Paterson, 2010). $\phi$ is a stiffening
factor estimated by solving an inverse problem (Cornford et al., 2015) using measured surface
velocities.





We use a viscous Weertman sliding relation to define the basal friction:
$$\tau^b = \begin{cases} -C|\boldsymbol{u}|^{m-1}\boldsymbol{u} & \text{if } \frac{\rho_i}{\rho_w}h > -b \\ 0 & \text{otherwise} \end{cases}, \ (7)$$
and here we assume a linear relation taking $m=1$. The basal traction coefficient C(x, y) is estimated
simultaneously with the stiffening factor $\phi$ by solving the inverse problem. $C$ and $\phi$ are adjusted
iteratively to reduce the misfit with a set of 2010 surface velocity observations (Joughin et al. 2010).
We hold the fields $C$ and $\phi$ constant over time throughout our simulations, although they must
actually change as the glacier retreats.

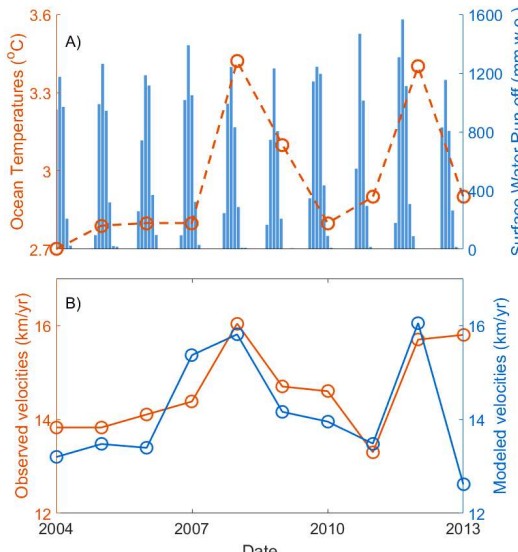


**Figure 2. A) Time series of ~300 m deep ocean temperature (red) from**
**http://ocean.ices.dk/HydChem/ near the mouth of Ilulissat fjord. Blue bars are simulated monthly**
**surface water run-off from the MAR regional surface mass and energy balance model (Alexander**
**et al. 2016). B) Measured ice front annual mean ice flow speeds (red) from Joughin et al. (2010),**
**compared with our modeled speeds (blue).**





## 2.2 Climate Forcing


We use ocean temperatures collected from a CTD site close to the mouth of Ilulissat fjord (Fig. 1)
as an approximation of ocean temperatures near the glacier grounding line. A comprehensive study
focusing on ocean circulation within Ilulissat fjord validated this approximation (Gladish et al.
2015). A positive correlation (r=0.74, p<0.05) exists between deep ocean temperatures and flow
speed near the terminus of Jakobshavn Isbrae (Fig. 2) from 2004 onwards. There is no significant
correlation prior to 2004, the floating ice tongue period. As a working hypothesis we assume that
the correlation since 2004 reflects the effects of the sea ice and iceberg mélange in the fjord on the
flow speed near the terminus: a warmer ocean reduces mélange thickness and therefore buttressing.
There appears to be no lag between the glacier acceleration and change in deep ocean temperature,
suggesting mélange response times are faster than 1 year. When the floating ice tongue was present
lags in the system were likely longer, accounting for the lack of correlation between ocean
temperatures and glacier flow speed prior to 2004. It is also possible that ocean temperatures reflect
changes in surface runoff and basal lubrication for sliding, but we consider that the runoff more
strongly affects calving mechanisms as discussed later. We therefore modify the driving force (Eq.
3) on the grid cells next to the calving front by multiplying by a factor $\alpha$ (tuned value shown by
Eq. 9 that is linearly related to ocean temperature (T) as a means of representing the buttressing
effects of the ice mélange in the fjord.
$$\nabla \cdot [\phi h \bar{\mu} (2\dot{\epsilon} + 2\mathrm{tr}(\dot{\epsilon})\mathbf{I})] + \boldsymbol{\tau}^b = \alpha \cdot \rho_i g h \nabla s, \quad (8)$$
$$\alpha = 0.82 + 0.111 \cdot T, \quad (9)$$
We use a crevasse based calving parameterization (Benn et al., 2007; Nick et al., 2011) that calves
ice where the crevasse penetration depth ($D_s$) is greater than upper surface elevation. $D_s$ is defined



as
$D_s = \frac{S}{g \cdot \rho_i} + \frac{\rho_w}{\rho_i} \cdot \spadesuit \cdot \beta,$  (10)
where $S$ is the magnitude of extensional stress, $\spadesuit$ is surface water run-off, and $\beta$ is a tuning scalar.
We estimate runoff from the 25 km resolution regional climate model, MAR, (Alexander et al. 2016),
driven by the ERA-Interim reanalysis (Dee et al., 2011).
We characterize submarine melting as a linear function of ocean forcing
$M_b = \gamma \, T_{f,},$         (11)
where $T_f$ is the far field ocean forcing temperature, taken in Disko Bay (CTD in Fig. 1), relative to
pressure melting temperature under the ice shelf. We derive $\gamma$ from the 1985 observed submarine
melt rate of $1 \pm 0.2$ m day$^{-1}$ beneath the floating ice mélange of Jakobshavn Isbræ, when Disko Bay
ocean temperatures were 4.2°C warmer than the pressure melting point at the bottom of the floating
ice shelf (Motyka et al. 2011). We test the sensitivity of the modeled glacier to uncertainty in
submarine melt rate in section 2.4.
We force Jakobshavn Isbræ in the 21st century using SMB and run-off from the 11 km resolution
RACMO model (Van Angelen et al., 2013) driven by the RCP4.5 scenario (Moss et al. 2010). The
run-off values are averaged over the nine grid points nearest to the terminus of Jakobshavn (69.1°N,
50.0°W). The RACMO simulation was forced by the HadGEM2-ES Earth system model (Collins
et al., 2011), as this climate model was found to be the most realistic for present-day simulations of
the Greenland ice sheet (Van Angelen et al., 2013). Ocean forcing should relate to temperatures off
the continental shelf close to the fjord mouth. Cowton et al. (2018) achieved success in simulating
the terminus position and yearly variability of 10 glaciers along the east coast of Greenland using



mean 200-400 m depth temperatures from reanalysis data. For consistency with the RACMO results,
we use deep ocean temperatures at ~ 300 m depth from the 0.83°×1° resolution HadGEM2-ES
driven by the RCP 4.5 climate scenario from 2005 to 2100 at the 3 closest grids point to Disko Bay.
We also compare this with results from 7 other climate model simulations of RCP4.5: HadGEM2-
ES (Collin et al., 2011), BNU-ESM (Ji et al., 2014), MIROC-ESM (Watanabe et al., 2011), IPSL-
CM5A-LR (Dufresne et al., 2013), CSIRO-Mk3L-1-2 (Gordon et al., 2002), NorESM1-M (Bentsen
et al., 2012) and MPI-ESM-LR (Giorgetta et al., 2013).

## 184    2.3 Initialization Procedure

As we are interested in high resolution simulations and validating our model parameterizations with
observations over the last 2 decades, we take care to initialize the model as accurately as possible.
Detailed bedrock topography and ice thickness data in the year 2009 come from Gogineni et al.
(2012). In 2004 the floating ice shelf disintegrated, making it a convenient starting point for
simulations since we might expect the system to respond differently to forcing when there was a
floating ice shelf compared with the situation of ocean forcing along a near-vertical ice cliff. This is
consistent with the observed good correlation between ocean temperature and flow speed after 2004
but not before. The aim of this initialization was provide a state rather similar to 2004, that is barely
retreating on inter-annual scales (Joughin et al., 2010) and small changes of annual mean velocity
in the following 3 years. Therefore





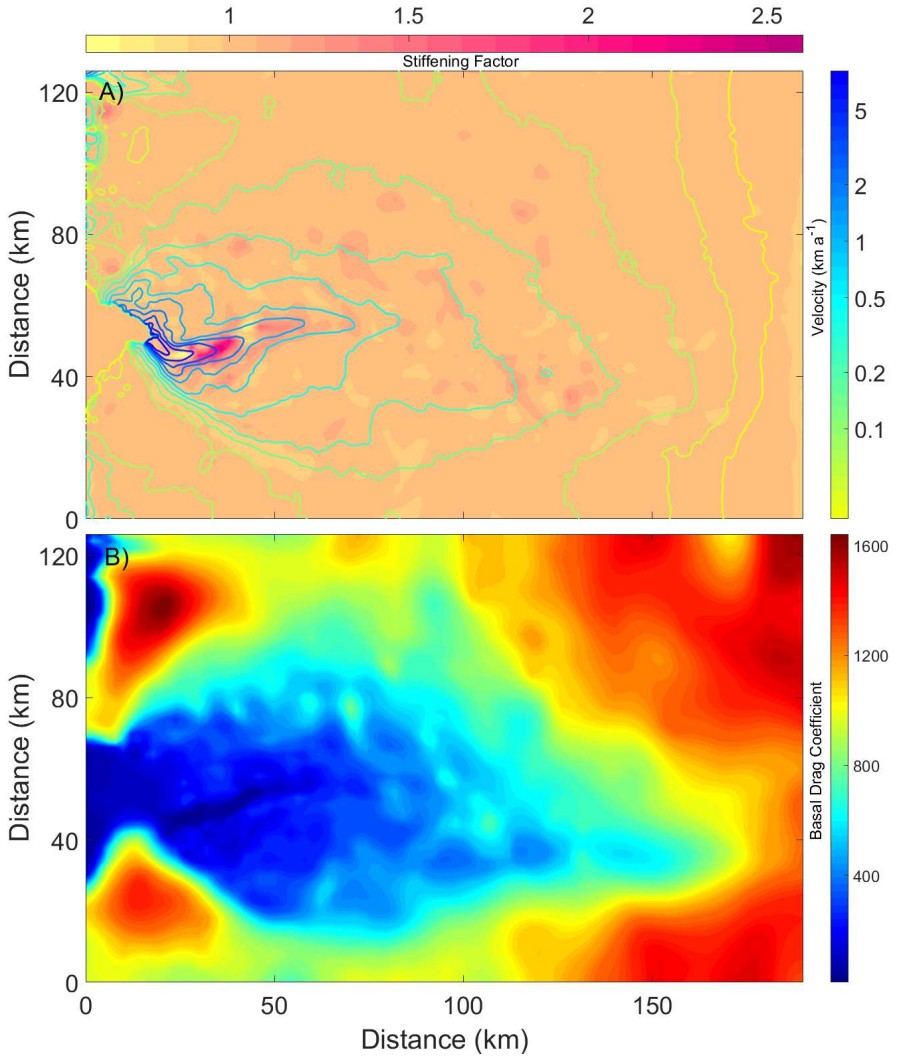

**Figure 3. (A) Stiffening factor $\Phi$ (Eq. 3) and (B) basal traction coefficient $C$ (Eq. 7) over the computational domain from solving the inverse problem. Contour lines in panel A show the modeled velocity (logarithmic scale).**

1) We solved the inverse problem for basal conditions (Eq. 7) and stiffening factor using 2010 velocities (Joughin et al., 2010) and 2009 geometry (Gogineni et al., 2012). Our friction



coefficient field is shown in Fig. 3.
2) We made an initial guess for β (Eq. 10) based on short (less than one decade) transient
simulations incorporating calving but not SMB or ice shelf melt forcing, starting from our
inverted model state from step 1 above. This was forced by seasonal repetitions of runoff
from MAR for the year 2010.
3) Starting from the inversion of step 1 and using β (Eq. 10) from step 2, we let the model
glacier evolve freely without calving and with zero SMB and with sub-shelf melting forced
by repeating the observed 2004 ocean temperature for 11 years (that means the coefficient
$\gamma$ in Eq. 11 was set to be 1) until its surface elevation profile was similar to the known
profile of 1998 (Bamber et al. 2001).
4) We carried out several 10-year simulations with $\beta$ gradually decreased from its initial value
given by step 2, to 45% of this value. These simulations were forced by repeatedly applying
the 2004 seasonal climate forcing. From these, we selected the $\beta$ that best allowed
Jakobshavn Isbræ to reach a stable state by the 8th year (changes in the 9th and 10th years
of simulations were negligible) such that its calving front position closest to that observed
in 2004. The best $\beta$ here is 53% of its value from step 2. This is our best guess for the 2004
state, but there are no thickness data available for 2004. Notice that the front positions and
May front velocities from 2004-2006 are stable (Figs 2 and 3), suggesting that the glacier
was reasonably close to steady state. This also makes 2004 a good time from which to start
transient simulations.
Basal friction coefficient values downstream of the 2010 grounding line were set equal to that
in the nearest 2010 grounded location. This was necessary because steps 2, 3 and 4 involved





grounding line advance beyond the region for which basal friction coefficients had been inferred.
The geometry after this spin up procedure, and the friction coefficient and stiffening factor
distribution from the inversion in step 1 were used as the initial condition for model calibration.

## 2.4 Model calibration

There are three essential parameters in the model, $\alpha$, $\beta$ and $\gamma$ representing mélange buttressing,
crevasse depth sensitivity to surface runoff, and shelf melt sensitivity to ocean temperatures. In the
initialization, we fix $\gamma$ to be 1. Therefore, we performed a suite of about 50 simulations to tune the
parameters $\alpha$ and $\beta$ with fixed $\gamma=1$. Our target was to best reproduce Jakobshavn Isbræ's calving
front position and surface velocity evolution for the 10 year period 2004-2013. Reproducing the
total distance of retreat and the temporary stable state after 2012 were secondary desirable features
to match.
Because only within a small range of $\beta$ will modeled retreats make sense, firstly we estimate
reasonable range of $\beta$ when hold $\alpha=1.0$, $\gamma=1.0$. Secondly, we explore the parameter space centered
by ($\alpha=1.0$, $\beta=0.06$), which come from estimations above, to match observed retreats and general
velocities neglecting the inter-annual variations. The parameter space tested here is ($1.0\pm0.25$,
$0.06\pm0.01$). As the discussion above, we further assume $\alpha = \alpha_1 + \alpha_2 T$, i.e., linearly related to deep
ocean temperature. With velocity depending on ocean temperature, degree of freedom of our
parameter space grow to three, which are $\alpha_1$, $\alpha_2$, $\beta$. However, we find $\beta$ and $\alpha_2$ behave quite
independently within the small $\beta$ range estimated so far, which allow us finally reach a set of
parameters that can accurately reproduce both the total retreats and the velocity variations including
inter-annual fluctuation. The tuning was implemented manually. The best set of parameters are





$\alpha_1$=0.82, $\alpha_2$=0.111 and $\beta$=0.0638, as shown in Eq. 9.
We explore the glacier's sensitivity to two types of boundary perturbations. They are ice mélange
buttressing effect (defined by $\alpha$) and submarine melting (defined by $\gamma$). We scaled submarine melt
rates by multiplying $\gamma$ by values from 0.8-1.2, based on the range of the observation uncertainty in
melt of ~ 20% (Motyka et al. 2011). Also we varied $\alpha$ by multiplying by factors from 0.91 to 1.25
to represent different buttressing strengths (Eq. 8). These multiplication factors were varied
systematically with typical intervals of 0.1 and 0.03 respectively for the $\gamma$ and $\alpha$ factors. We
calculated the following relative mismatches defined as (model-observations)/observations for each
simulation (shown in Fig 4):

254        1.  Total calving front retreat from 2004-2013

255        2.  Annual mean front velocities

256        3.  Vector sum of 1) and 2)

We used $\beta$ (Eq. 10) from our optimal set of parameters. Our optimal value for $\alpha$ is such that a
20% rise of its value does not affect modeled retreat when $\beta$ and $\gamma$ are kept to be their optimal
values (Fig. 4 A).





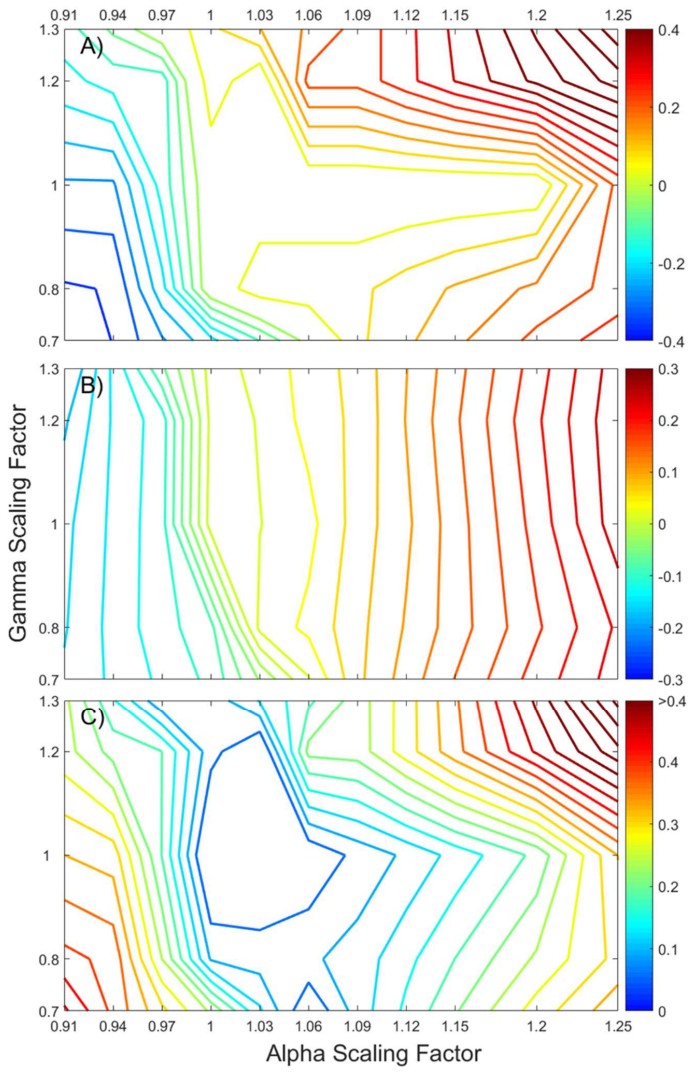

**Figure 4. Relative mismatches defined as (model-observed)/observed for A) total calving front retreat, B) average of annual mean front velocity during 2004-2013, C) the vector sum of mismatches in panels A and B, $\sqrt{(A^2+B^2)}$ in our 2-D parameter space $(\alpha, \gamma)$ centered by the best set $(\alpha * 1.0, \gamma * 1.0)$. X- and y-axis are multiplier of $\alpha$ and $\gamma$ used respectively in different runs.**





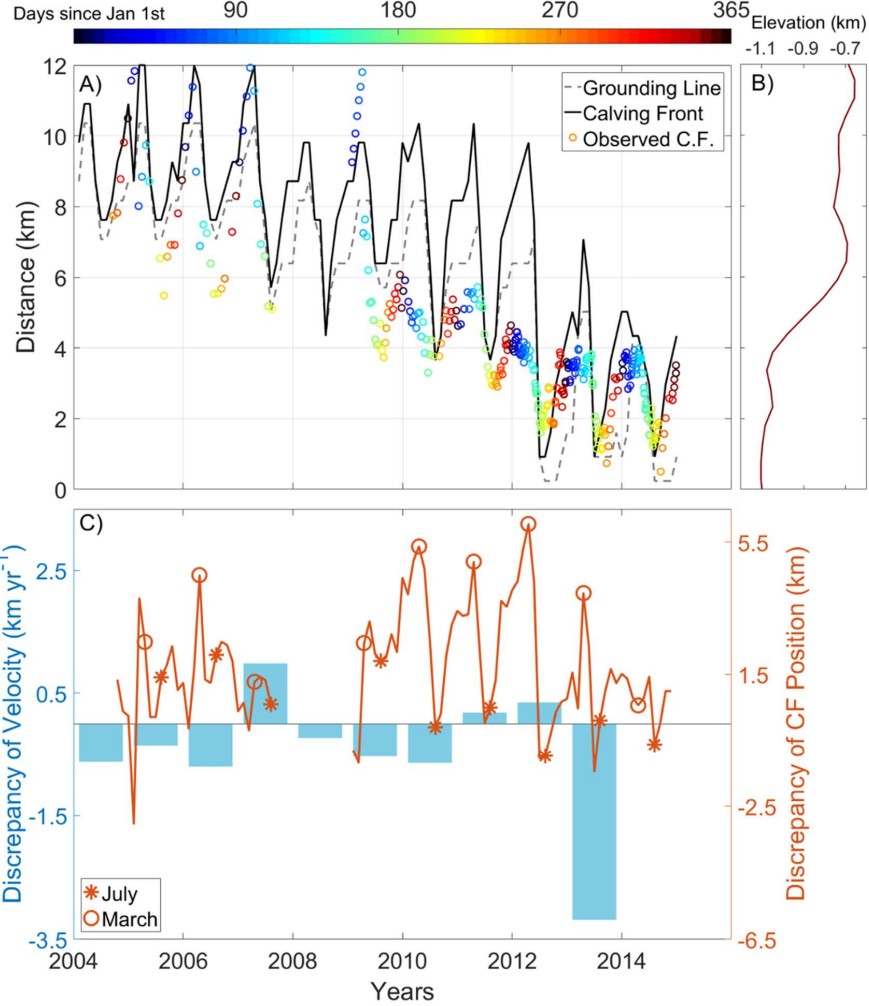

266

**Figure 5. (A) Modeled retreat of the calving front (black solid line), grounding line (gray**

**dashed line), and observed calving front positions (color-coded circles and scale bar) from**

**Joughin et al. (2014). (B) Bedrock elevations. (C) Residuals (modeled minus observed) of**

**annual mean front velocity (blue bars, left axis) and of calving front position (red lines, right**

**axis) with typical timings of annual maximum (March) and minimum (July) extent marked.**

**The modeled front velocities and calving positions explain about 49% and 76% of the variance**





**in corresponding observations.**
The two biggest mismatches occur with the 2007 and especially 2013 velocities (Fig. 5). 2013 has
the lowest simulated surface water run-off (Fig. 2) of all the years since 2004. The Benn calving
model we use is sensitive to runoff, with reduced run-off leading to lower crevasse-penetration-
depth and reduced terminus fracturing thus increasing its buttressing force. In the 2012/13 winter,
the modeled glacier had an unprecedentedly high calving front and had been flowing fast the
previous summer. This led to growth of an unusually long seasonal ice-shelf in the winter which
caused low velocities near the end of front advancing season, and so accordingly low annual mean
velocity. The low modeled velocity in 2009 can also be interpreted by the same over-growth effect,
even though the ocean temperatures were high in 2009. Jakobshavn Isbræ did not in fact slow down
very much in 2009 and 2013 because there were calving events that are unrepresented in our model.
The relevant mechanisms are discussed later.
In 2007 high run-off caused more simulated calving and retreat than in reality. These retreat phases
reduced the buttressing and lateral drag due to shear-margin-weakening, all of which lead to
excessive speed-up near the terminus.
Modeled calving front retreat is ~ 7 km in total from 2004-2014 (Fig. 5), which is consistent with
observations (Joughin et al. 2014). In 2009 a dramatic retreat brought the grounding line to the
bottom of the bedrock slope, and since then it has gradually retreated with smaller seasonal
fluctuations. The run-off forcing we applied triggered major retreats in the summers of 2007 and
2012, due to large summer peak run-off (Fig. 2), demonstrating the sensitivity of our calving
parameterization to run-off forcing. Modeled timings of maximum extent and minimum extent each
year are in good agreement with observations, also demonstrating that summer, in particular, May
to July, run-off determines much of the behavior of Jakobshavn Isbræ.





The modeled range of seasonal fluctuation in front position is ~ 5 km, which is similar to
observations in the period before 2008. From January 2009 to December 2011, there was an abrupt
decrease in seasonal front fluctuation, with many winter calving events occurring, in contrast with
previous years (Cassotto et al. 2015). These winter calving events may explain the small observed
seasonal fluctuations because they limit the winter advance. Our model is unable to stimulate these
winter calving because there is no winter run-off, and hence calving is zero then. The largest
discrepancy of front position occurs during these winter calving periods (Fig. 5). Observations also
showed that from 2013 to 2017, Jakobshavn Isbræ barely retreated (Joughin et al. 2010). The decline
of run-off (Fig. 2) in 2014 suggests the reason. But since no RACMO run-off simulations are yet
available for 2015 and later, our parameterizations cannot be tested against this lack of retreat.
# 3 Future evolution

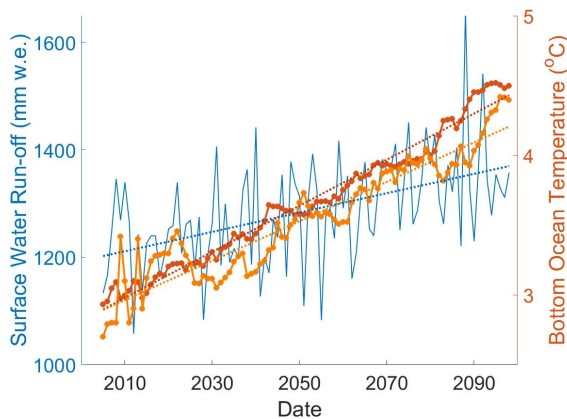

**Figure 6. Climate forcing for future projection under the RCP4.5 scenario taken as 300 m**
**depth ocean temperatures from HadGEM2-ES (orange) compared with the ensemble mean**
**(red) of 7 Earth System Models (HadGEM2-ES, BNU-ESM, MIROC-ESM, IPSL-CM5A-LR,**
**CSIRO-Mk3L-1-2, NorESM1-M and MPI-ESM-LR), (right axis), with their linear trends.**
**Annual maximum monthly surface water run-off near Jakobshavn Isbrae's terminus from**
**RACMO is shown in blue.**

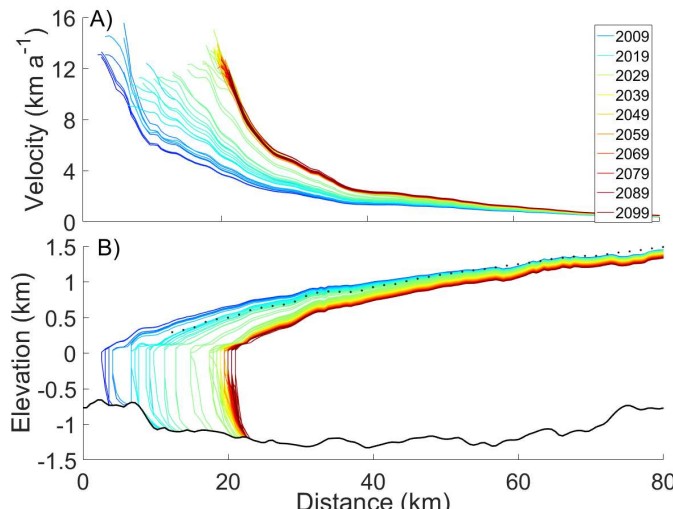

**Figure 7. Modeled profiles of (A) January velocity and (B) January surface elevation along**
**the center-flow-line of Jakobshavn Isbræ from 2004 to 2099 for the RCP4.5 scenario. Bedrock**
**elevation is shown in black. Black dotted line is the surface elevation profile extracted from**
**radar data measured around 2010 (Gogineni et al., 2012). Profiles are shown at intervals of 1**
**years. Profiles are color-coded in the legend and range from blue to green and red.**
Under the RCP4.5 scenario (Fig. 6) surface runoff slowly rises over the 21st century, with RACMO
simulating slightly greater runoff during the second half than for the first 50 years. Runoff increases
by 14% over the century. Bottom ocean temperature in the grid cell closest to Jacobshavn increases
by 52%, and, as may be expected, has less variability than runoff.
Under this forcing, Jakobshavn Isbræ continues its retreat (Fig. 7) for 18 years after 2013, producing
a total grounding line retreat of ~18 km upstream. As calving produces a steepening surface profile,
terminus velocities increase, to reach a 21st century peak of ~19 km a$^{-1}$ in 2031 summer. Eventually
the calving front becomes higher than the crevasse penetration depth in the calving parameterization.



This leads to a stable period with little inter-annual retreat and which lasts until the end of this
century. During this period, nearly all of the seasonal retreats are offset by the following winter re-
advances. Mass transport continually flattens and thins the ice geometry, leading to reduced flow
speeds that eventually become half those of 2031, the 21st century peak.
The surprisingly high run-off anomaly in 2088 (Fig. 6) does not affect the stable state indicating
run-off fluctuation alone cannot break this retreat pattern immediately. Once the inter-annual retreats
cease in 2031, the dynamic thinning rate is greatly reduced because calving front height stops
increasing.
**Table 1 Estimates of glacier mass loss and grounding line retreat from different sources.**

| Source | Climate scenario | Mass loss 2004-2013 (10 years) (Gt) | Mass loss by 2100 (Gt) | Grounding line retreat 2004-2013 (km) | Grounding line retreat by 2100 (km) |
|---|---|---|---|---|---|
| This paper | RCP4.5 | 234 | 2068 (2044-2723) | 7.0 | 18.5 (17.5-23.0) |
| Muresan et al. (2016) | | 220 | | | |
| Nick et al. (2011) | A1B | | 1870 - 2281 | | 14.0 - 26.0 |
| Observations | | 225±15 | | 7.0 | |

Table 1 shows estimates of glacier mass loss and retreat. Under RCP4.5, total cumulative mass
change of Jakobshavn Isbræ is 2029 Gt by 2100, using best set of α, β and γ with ocean temperature
inputs from ensemble mean of 7 ESMs (Fig. 6). To estimate an upper bound for mass loss over this
century, we scale the α parameter by 1.2 giving 2680 Gt for the same forcing. Using the HadGEM2-
ES forcing, which is the same model used to force RACMO with α and γ set to their best estimates
(Fig. 4) gives 2000 Gt. We suggest that this may be the lower reasonable bound of mass loss since
the HadGEM-ES ocean temperatures rise notably slower than the ensemble mean.
Exploring the (α, γ) parameter space we notice that values of (1.0, 0.8) produces a mass loss over
this century of 2021 Gt with the HadGEM-ES ocean forcing, almost the same value as for the best



set of parameters. This implies that less submarine melting (determined by γ) leads to larger ice loss
by dynamic processes. The reason is that lesser submarine melt allows a larger ice thickness at the
grounding line with stronger dynamic thinning in advancing season. Notice in our stress balance
equation (Eq. 3), thickness contributes to driving force term, thus ice flux across the grounding line
is highly nonlinear in ice thickness. This highly nonlinear relationship is also shown in our
sensitivity tests (Fig. 4). Over the mismatch field measured by front velocity (Fig. 4, Panel B), the
velocity is partly dominated by low values of γ around the line α = 1.06, while α is almost the only
control on velocity over the region where α<1.09. Within our sample space, the non-linear and non-
monotonic relationship between submarine melting and retreats is clear (Fig. 4, Panel A). Around
the point (α = 1.12, γ = 1.0), total retreat will increase no matter if γ is decreasing or increasing
within the range 0.8 < γ < 1.2. The area α > 1.0 in sample space is the very likely future condition
for Jakobshavn Isbræ because increasing terminal ice cliff height caused by retreating into deep
water will act as an amplifier to frontal driving force.

# 4 Discussion

## 4.1 Parameterization of Buttressing effect

The sudden 1.1℃ rise in temperature of water entering Ilulissat fjord in 1997 (Holland et al., 2008)
initiated rapid melting and disintegration of the floating ice mélange in 2003. This disintegration
coincided with a near doubling of ice velocities. Modeling (Vieli et al., 2011) suggested that this
was due to the reduction in buttressing from the floating ice-mélange. We can realistically reproduce
the velocity variation of Jakobshavn Isbræ on seasonal and inter-annual scales using our
parameterization of the buttressing effect from the ice mélange in the fjord.



Gladish et al. (2015) analyzed glacial flow speeds from 1998 to 2014, finding no correlation with
Ilulissat fjord temperatures. This is because at the beginning of 2004, Jakobshavn's evolution entered
a new phase with the disintegration of the ice mélange and floating ice shelf. We find good
correlations between Disko Bay temperatures and ice velocities from 2004 to 2014. The
improvement in correlation with temperatures may be explained by a faster response between the
grounded glacier and the fjord water temperatures after loss of the floating ice shelf. Thus only
freshly calved icebergs played roles in providing terminus resistance, and these could be reasonably
supposed to react to seasonal fjord temperatures very quickly.
Buttressing would affect the calving process by altering the longitudinal resistive stress in the glacier.
Temperatures in Ilulissat Fjord will be warmer during the 21$^{st}$ century under essentially all climate
scenarios, even those with modest emissions, due to the thermal inertia of the oceans. Thus a new
floating ice shelf is unlikely to form. Prior to 2004, there were large changes in Jakobshavn: loss of
~15 km long stiff ice mélange and the sudden rise in fjord temperatures in 1998. There are fewer
mechanisms to effect such dramatic changes in the future now that almost the entirety of the glacier
is grounded. We therefore propose that our representation of the mélange buttressing mechanism,
tuned for 2004-2013, is likely to maintain its validity during the 21$^{st}$ century.

## 381   4.2 Horizontal shearing and viscosity

Van Der Veen et al. (2011) estimated a maximum horizontal shear stress of ~800 kPa across the
shear margin of Jakobshavn Isbræ where the horizontal velocity shear reaches the peak, while the
bed stress is only 10-40 kPa in fast flowing regions (Shapero et al., 2016). Given that the width of
the Jakobshavn Isbræ fast flow region is typically under 5 km and its thickness is typically between
1-2 km, these numbers indicate that the shear margins provide at least an order of magnitude greater
total resistance than the bed. Thus, the shear margin, rather than the bed of Jakobshavn Isbræ





provides most of the resistance balancing the driving force. The main trunk of Jakobshavn Isbræ
exhibits considerable seasonal velocity changes, while the slow moving ice outside the shear margin
has little or no seasonal cycle. This flow structure implies speed gradients perpendicular to the flow
direction with large seasonal variation. These velocity shears would in turn generate large seasonal
variations in effective ice viscosity (Eq. 6). This mechanism implies a positive feedback on velocity
in the fast flow region: increases in the speed of fast flowing ice cause increases in horizontal shear
stress across the margins, reduced viscosity, and further increased horizontal velocity shear,
allowing further increase to speeds in the fast flow region. Observations show that, as the terminus
retreated into deeper water, seasonal fluctuations in terminus velocity increased (Joughin et al. 2008).
By 2012, the summer time peak terminus velocity was ~ 17 km a$^{-1}$, more than twice the wintertime
minimum velocity (Joughin et al. 2014). This amplified seasonal velocity cycle was likely enhanced
by the shear-margin weakening mechanism.

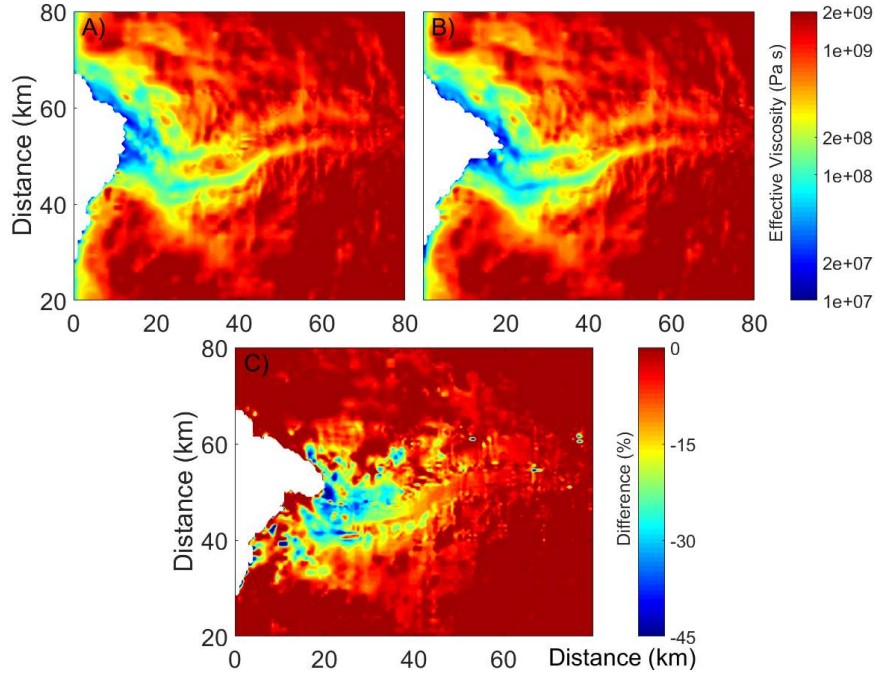






**Figure 8. Modeled annual mean of vertically averaged effective viscosity $\Phi\mu$ (Eq. 5) in 2004**
**(A) and 2013 (B) and the percentage decreases from 2004 to 2013 (C).**
Our modeled shear margin weakening on decadal scales is consistent with other estimates from a
thermomechanical ice flow model of Jakobshavn Isbræ forced by calving front positions (Bondzio
et al., 2017). Their modeled viscosity drops between 2003 to 2015 reach ~ 40% which is close to
our maximum viscosity decrease of ~ 45% between 2004 to 2013 (Fig. 8). The extreme calving
season we simulated in summer 2012 was accompanied by ~ 12 km a$^{-1}$ variations in speed at the
calving front, which were facilitated by the accompanying shear margin-induced ice viscosity
reductions of 60% at the time of maximum terminus advance. Simpler models of Jakobshavn Isbræ,
using a flowband model (Nick et al., 2011) or simple calving parameterizations with no seasonal
cycle (Muresan et al., 2016) cannot produce these seasonal variations in shearing. However, our
model accommodates both the seasonal forcing from calving and the three-dimensional seasonal
velocity shear impacts on effective viscosity. Without this physical process, speedups during intense
calving events would be under-estimated, and this would lead to under-estimated mass
transportation during the retreat.
## 4.3 Comparison with previous estimates
The cumulative mass change of Jakobshavn Isbræ estimated from airborne and satellite laser
altimetry for 1997–2014 was tabulated Muresan et al. (2016). The mass loss over the 10-year period
2004-2013 modeled by Muresan et al. (2016) is closer to observations than ours (Table 1). This is
partly due to different tuning targets: matching observed mass change was a stated target in their
study, whereas our study targets ice front position and velocity. Their close match to observed mass
loss may be partly due to cancelling errors: 1) their modeled calving front barely moves after 2006,
which leads to under-estimation of mass change; and 2) the modeled fast flow widths are larger than



observations, which amplifies the mass flux across the calving front. These two biases will not
always offset each other perfectly in the future.
Muresan et al. (2016) failed to simulate the retreat of Jakobshavn Isbræ after 2010. This may be due
to the thickness threshold employed in their calving parameterization. Once Jakobshavn Isbræ
terminus has retreated into the deeper part of the bedrock trough, the terminus height might never
drop below their calving threshold of 375 m. In this case their calving rate will be solely due to the
eigen parameterization of strain rates. Moreover, absence of seasonality in their calving front leads
to under-estimated dynamic thinning, which is a key prerequisite for further calving. In contrast,
our crevasse-depth calving model depends on stresses and surface water run-off with strong seasonal
variation. As the terminus retreats and the surface slope steepens the enhanced surface stretching
enhances the opening of crevasses in both calving parameterizations.
Nick et al. (2011) used a flow-band model to estimate a mass loss of 2280 Gt for Jakobshavn Isbræ
by 2100 under the A1B climate scenario (Table 1). In our model we use RCP4.5 climate forcing,
which has lower temperature rises than A1B, especially after 2050. Nick et al. (2011) prescribed a
flow-band that has a near uniform width of 5 km near the terminus. Later modeling work using a
similar model suggested that stability of the glacier is fundamentally controlled by geometry, and in
reality the width varies along the ice-stream (Steiger et al. 2017). Nick et al. (2011) chose sets of
parameters that produced small inter-annual retreats of Jakobshavn from 2000-2010, which may
limit mass loss and retreat. The absence of the shear margin weakening feedback in their model also
likely causes underestimation of mass loss. This could account for the comparable projected mass
loss to our results, and less terminus retreat (Table 1), even though their climate forcing scenario
was warmer.





## 4.4 Model improvements

We overestimate mass loss relative to observations over Jakobshavn Isbræ drainage basin for 2004-
2013 (Table 1). The main reason is excessive dynamic thinning over the lowest $\sim 20$ km of the main
trunk due to over-estimated summer speed. For example, modeled front velocity soared to a peak
of $\sim 20$ km $a^{-1}$ in summer 2012, while the observed maximum speed is only 18 km $a^{-1}$ (Joughin et
al., 2014). In this summer, we simulated a series of full-thickness calving that eventually left an
unprecedented tall ice cliff. In reality, calving events do not always occur to full thickness, thus the
glacier tends to form a shorter ice cliff that caters for lower velocity and less dynamic thinning.

Since the grounding line of Jakobshavn retreated to the bottom of a reverse bed slope in 2009, the
height of the calving front has generally increased, causing larger mass flux downstream across the
calving front. Instead of enhancing the seasonal fluctuation of calving front position, substantial
winter calving events have occurred instead. Given the fact that these calving events have reduced
the typical winter advance from $\sim 6$ km to $\sim 3$ km since 2010, winter calving is now likely as
important as summer run-off-driven calving. During this period of low magnitude seasonal
fluctuations, a series of retreats gradually moved the calving front position on inter-annual scale. In
contrast, the inter-annual retreats before 2009 were mostly driven by single calving seasons, e.g.,
May to July 2009. Our model using the Benn calving model is better able to stimulate this earlier
retreat pattern, which is largely determined by each year's peak surface water run-off.

The grounding line of Jakobshavn Isbræ is unlikely to return to shallow water in the remainder of
the 21$^{st}$ century because bedrock elevations $< -1000$ m beneath the main trunk further extend a
further $\sim 60$ km inland. Accordingly, the latest retreat pattern including winter calving, is likely
closer to the pattern of future evolution of Jakobshavn Isbræ. A short floating part due to winter
calving is always accompanied by weaker lateral drag and steeper surface slope near the grounding



line, all of which are conducive for faster ice-flow. So, winter calving would enhance the
downstream mass transportation, a missing process in our model.
The process of winter calving must take place without any surface water. That calving must be
generated by processes affecting ice cliff stability, and that is likely due to changes at the base rather
than the surface. Evidence of calving by opening of basal crevasses and splitting comes from
terrestrial radar showing the terminus lifting several days prior to a large calving (Xie et al., 2016;
James et al., 2014). These observations suggest that the glacier is not in hydrostatic equilibrium
during calving. Our simulation specifies the glacier is in hydrostatic equilibrium on timescales of
the simulation. Our model cannot simulate the process of up-lifting. Instead we assume the upper
and lower surface would instantly lift to the state of floating (Eq. 1). However, there is some
evidence that Jakobshavn must behave super-buoyantly in winter. We observe that the simulated
grounding line of Jakobshavn retreats even after cessation of calving front retreat (Fig. 3). These
retreats can be explained by rapid dynamic thinning near the grounding line leading to its buoyancy
exceeding gravity and, consequently, floating.
A combination of discrete element model and continuum ice-dynamic model (solving the 3-
Dimensional full-stokes equation) is able to reliably replicate observed calving styles in the case of
a super-buoyant terminus (Benn et al. 2017). The discrete element model allows investigation of
calving processes in unprecedented detail by analyzing the stress pattern dominated by glacier
geometry and boundary conditions. However, these calving processes are beyond the capability of
calving parameterization based on surface crevasse depth assuming depth-independent flow. Better
understanding of this buoyancy-driven calving and further model development to represent more
details such as fracture propagation are needed to accurately simulate glacier's future evolution.





# 5 Conclusion


We use a three-dimensional dynamic ice-sheet model with a physically-based calving
parameterization to model the evolution of Jakobshavn Isbræ. After tuning the parameters, our
model can accurately reproduce Jakobshavn Isbræ's retreats and velocity changes from 2004-2013
on both seasonal and inter-annual scale. We project Jakobshavn Isbræ's future dynamic changes
with climate forcing data from RACMO (2014-2099) and an ensemble of 7 Earth System Models
for the RCP4.5 scenario.
We successfully model two-dimensional ice-flow patterns and their seasonal variations for
Jakobshavn Isbræ, which are missing from several previous modeling studies. Moreover, capturing
these two-dimensional patterns allows us to handle the influence of horizontal velocity shear on
effective ice viscosity, which impacts on speedup processes of Jakobshavn Isbræ.
Over most of the 21st century, Jakobshavn Isbræ's grounding line will, we predict, retreat along the
deep parts of a basal trough where bedrock elevation is significantly lower than at the present
grounding line. Using the current generation of calving parameterizations, which are essentially
thickness threshold models, is challenging because of the increasing height of the calving front as
Jakobshavn Isbræ retreats, meaning that crevasse penetration depths become too small to initiate
calving. Our model successfully reproduced Jakobshavn Isbræ's retreat down a reverse bed slope
with an elevation drop of ~ 400 m and the subsequent temporarily stable calving front position in
2013 and 2014.
Our results suggest that rapid dynamic thinning and calving caused by deep crevasse penetration
are responsible for most of its recent mass loss, and will be a decisive process in future mass loss.
Further exploration of the physics of calving and basal sliding of Greenland outlet glaciers are

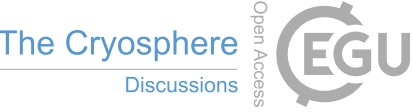


required to improve future projections.
**Acknowledgements**
This study is supported by National Key Science Program for Global Change Research
(2015CB953601), National Key Research and Development Program of China (2018YFC1406104)
and National Natural Science Foundation of China (No. 41506212). We thank Stephen Cornford for
his help in implementing some parameterizations used in our model.

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
