# Peer review of "Simulated retreat of Jakobshavn Isbræ during the 21st"

_The Cryosphere, 2019_

## Referee Comment (RC1) · Anonymous Referee #1 · 22 Feb 2019

This paper presents a modelling study of Jakobshavn Isbrae using the ice flow model BISICLES. The model is initialised and calibrated to fit the observed front retreat and annual velocities between 2004 and 2013.

Three parametrisations are used to control the position of the front:

- basal melting (Eq. 11),

- calving based on a crevasse depth criteria (Eq. 10),

- and a parametrisation meant to represent the buttressing of the ice melange in the front and that affect the driving stress (Eqs. 8-9) near the front.

[Figure]

Because it is a target for the calibration, the total observed front retreat is well repro-duced by the model. The timing of the front variations with winter advance and summer retreat is captured by the model, however the seasonal variability is overestimated at the end of the period because the model does not reproduce the winter calving. Finally, the model is used to estimate the evolution during the 21st century.

The results during the calibration are convincing and well discussed. However, I found that few points are missing for the description of the model and set-up, the initialisation and calibration is relatively hard to follow as it involves many steps that have been implemented manually and the discussion mainly concentrates on what the model has and forget to include what is missing. I give more details below.

- My main remark concerns the parametrisation of the buttressing by the ice melange. The boundary condition for the front is not mentioned in the model description but should be the difference between the force exerted by the ice and the back stress from the sea water. It would seem natural to implement the effect of the ice melange as an additional back stress. The parametrisation im-plemented here modifies the driving stress near the front. More justification for this implementation is really required. Is this process really needed to reproduce the front variations? What appends if there is only the calving parametrisation? Moreover, the effect is proportional to the fjord temperature and is thus con-tinuously increasing in the future simulations. However, as the temperature in-creases, we may expect some kind of threshold where the ice melange disap-pears and its effects become negligible?

- Few important informations are missing for the model description: What is the temperature field for the initialisation? Is the model thermo-mechanically cou-pled? What is the mesh resolution?

- The description of the initialisation is very hard to follow. For example for the step 2, we don't know what is the target to adjust $\beta$. In step 3, it is said that $\beta$ from

step 2 is used but that there is no calving, however $\beta$ controls the calving criteria. It is said that $\gamma = 1$, however in page 9, Eq. 11, it is said that $\gamma$ was derived from the 1985 observed submarine melt rate.

- The discussion mainly focuses on the effect of the shear margins and the fact that due to the non linearity in the ice flow law the effective viscosity decreases as the strain-rates increase. This mechanism is described as a positive feedback, however I'm not sure that this is the right term, as both the velocities and the strain-rates are the results of the force balance equation, so that the velocities and the effective viscosity depends on the model parameters and boundary conditions. But is it difficult to describe this as a feedback as they both are solution of the same equation. However, the results are certainly dependent of the ice flow law and the value of the stress exponent, and we certainly may expect that the results would be different with a linear flow relation, or, as discussed, a flow band model that would parametrised the lateral drag. The mesh resolution might also influence the results as the resolution should be sufficient to properly capture the steep velocity gradients to represent this effect. In addition, the comparison with Bondzio et al. (2017) might be a bit confusing as Bondzio et al. include the thermo-mechanical coupling (which is absent here?), and they report that the warming from shear heating accounts for 20 to 30% of the decrease in effective viscosity. There is also several mechanisms that could affect the viscosity of the shear margins with potential feedbacks, this includes damage, cryo-hydrologic warming, anisotropy etc... This could be discussed also.

- Finally it would be also interesting to see or at least discuss how the model results are sensitive to other uncertainties in the model; this includes the description of the bedrock and the basal friction law, especially the linear assumption that is used here.

---

## Referee Comment (RC2) · Signe Hillerup Larsen (Referee) · 25 Feb 2019

General comments:

The model approach resembles previous studies by Muresan et al. 2016 by using ocean temperatures as a forcing to a dynamic ice flow model. However, Xiaoran Guo et al. expands the approach by going into more detail on seasonality and viscosity changes, while also starting their model in 2004 (not in 1990 as Muresan et al does) where they provide evidence that there is a shift in flow regime. Thus, there is a scientific advance within the field, by exploring ways to improve methods for modelling the behaviour of fast flowing ice streams.

These types of model studies requires a lot of technical settings and tuning of the model

which is very complicated and hard to explain in an easy-to-understand way. However, in order to satisfy the demand of traceability of results, this is the most important part of the paper. The model setup sections are not doing this sufficiently, in their current state.

Specific comments:

Model description sections: Initialisation and calibration should be improved to make it clearer exactly what has been done. In particular I am missing information about what basal and surface geometry is used in the inversion process and also how values for basal friction and ice softness are derived. Furthermore, I am curios about the mesh resolution used in the model and in particular how this looks across the shear margins.

Basal geometry: It is not stated anywhere what basal geometry is used. As, the authors also state in the discussion, geometry is the most important factor for ice stream stability and thus the results of the retreating calving front should furthermore be mapped on top of a basal geometry map in 2d plan view (seen from above). The retreat pattern relation to basal geometry should be discussed in relation to other studies modelling the future behaviour of Jakobshavns Isbræ.

In relation to starting in 2014: To my understanding, and also what you describe for the model, a stiff ice mélange has a butressing effect. Thus, it seems strange to me that the glacier is stable from 2004 and onwards, if it just lost an important butressing?

Line by line comments: Section 1 Generally, there is confusion about the definition of a floating ice shelf and a stiff ice mélange throughout the section.

Line 70-72: Needs a reference

Section 2 Line 102: what basal map do you use? Line 123-124: Please refer us to a study where the method of solving the inverse problem where two unknown is discussed (or explain in detail here how that would work, and how you can trust the outcome). I think this is an important point as viscosity is non-linear.

[Figure]

Section 2.2: Should just be titled Forcing (and not climate forcing) Line 135: What is CTD? Line 136: At what depth is the ocean temperature a good approximation? Line 153: Use alpha1 and alpha2 in stead of the calibrated numbers Line 157-160: MAR is used to estimate the runoff in equation 10. Later on Racmo is used as forcing. It is not clear why you use two different models, and when they are used. Line 169: Make it clear that it is your model your are talking about Line 169: Write out SMB Line 174-177: Please state in what equation this ocean forcing goes into

Section 2.3

Line 187: The dataset described here is only 2d, your model is in 3d, so I am not sure what you are using this for? Line 188: to my understanding, the sudden disappearance Line 202: please remind us what beta is Line 209-210: What does similar mean? How far off are we talking here? And please state why you use the 1998 profile when the model is starting in 2004. Line 214: Why is it the 8th, needs clarification. Line 217: Aha, good to know already in line 209-210 Line 218-219: The glacier is definitely not in steady state in 2004, please rephrase 2.4 Model calibration This section is very confusing to me. I think it needs a rewrite to become clearer.

Line 235: rephrase sentence Line 235-245: I am confused about this whole paragraph. The following paragraph (Line 246-259) is better structured, could this perhaps be the start of the section? Line 274-284: This whole paragraph needs clarification.

Figure 5A: How is the calving front retreat defined? Is it just a comparison at points along a center flowline? And it this representative of the general retreat?

Section 3 Figure 7: I would be more interested in seeing the retreat from above, the center flowline bedmap does not explain the stop of retreat.

Line 322: Make it clear that you are talking about you model version of Jakobshavn Isbræ

Section 4 It confused me a bit that you called this Discussion as a lot of important

results are also presented here.

Line 370-372: This sentence does not make sense to me, does freshly calved ice bergs really provide any resistance?

Line: 376-377: here you call it a stiff ice mélange, I think you should use this term throughout, especially in the intro

Section 4.2 I am a bit confused, are the results of changes in the effective viscosity shown in figure 8 results from your forward run? And if so, how does the fact that you are keeping ice softness constant influence these results? I think there must be an effect in the softening from the thermodynamics as well?

Section 4.3 Good to have comparisons with previous results, I think a key point, which you focus very little on, is that the retreat stops in the same area in all the studies (if I understood this correctly)? I think that if you also add figures showing basal geometry and retreat as suggested earlier, this point is easily added.

Line 498-499: What do you mean by two-dimensional ice flow patterns?

---

## Referee Comment (RC3) · Anonymous Referee #3 · 5 Mar 2019

**1   Summary statement**

The manuscript "Simulated retreat of Jakobshavn Isbrae during the 21st century" by X. Guo and colleagues presents results on the simulation of Jakobshavn Isbrae over the 21st century, calibrated to match its current configuration and recent evolution. The model includes buttressing provided by the ice melange and a calving law based on crevasse-depth, and the forcings are based on Global Climate Models (GCMs). The results suggest that the glacier will continue to retreat and lose mass during the 21st century, reaching 5.6 mm of sea level equivalent by the end of the century.

The paper is well written, usually easy to follow (except for the initialization procedure that is quite complicated), and the figures appropriate. However, some additional ex-

planations are needed to understand the choices made for the calibration of several parameters, for some of the datasets used, or for the initialization procedure. Furthermore, only a couple of figures show the evolution of the glacier over a flow line for one given simulation of the ensemble. It would be valuable to show the spread of the model results for the different parameters used and the different forcings, but also to show the spatial evolution of the ice front not just on a flow line but for the entire basin. Finally, the authors mention that the calving law based on crevasse-depth prevents the calving of the glacier once the thickness becomes too large. This is the contrary of what is physically expected: a tall cliff with a large height above sea level leads to more calving, so there is no reason for the calving to get reduced towards the end of the simulations.

**2  Major comments**

The bedrock and bathymetry used come from Jakobsson et al. (2012) and Gogineni et al. (2012), while the newer bedrock elevation maps of Greenland typically used in ice sheet modeling are Bamber et al. (2013) and Morlighem et al. (2017), so it is a rather interesting choice. There are probably good reasons for using these maps, but they are not well explained. It would be good quantify the impact of this choice on the simulations compared to other choices, or at least explain the differences expected.

The calving law is based on crevasse-depth based calving only, so that calving happens when deep surface crevasses develop in the presence of surface water. Is this representation of calving sufficient to represent the different types of calving throughout the year and as the glacier retreats to deeper grounds? It seems that the model is not able to simulate calving in winter that is becoming important towards the end of the simulation. Would a different parameterization of calving lead to different results? This is a rather important question as it contradicts the marine ice cliff instability that

predicts faster and faster retreat as glaciers retreat to deeper grounds and the ice thickness increases. So what is the impact of choosing a crevasse-depth based calving? This should be addressed in more details in the discussion.

The role of melange and its parameterization are said to have a relatively large impact on the results, but the exact role of melange and the associated processes that could impact calving remains unclear. What happens to the simulations when the buttressing provided by the ice melange is removed?

The processes included in the simulations include many parameters, and these parameters are not always justified or explained. In particular what they physically represent and what the impact is for the simulations. For example: How much buttressing does the melange represent? What is the equivalent ice thickness needed to get a similar buttressing? What is the tuning scalar for the run-off in the crevasses? What ocean temperatures are used for the forcing, and how was this choice made? How are the ocean temperature converted from the far field, to the fjord and to the grounding line region?

The initialization is rather confusing, with a target date of 2004 at the beginning of the simulations, but other datasets with different times are used for the inverse problem (2012) and the relaxed surface elevation (1998). This part should be clarified to better understand the rationale behind the initialization procedure.

I am wondering how reliable the GCMs are to reproduce the temporal patterns of variability of the glacier: to my knowledge, GCMs do not get the right timing for the variability, so maybe some reanalysis data would perform better for that.

Finally, there are not many figures showing the results, e.g., the spatial distribution of ice front position at the end of the simulations for the different cases or the mass loss for the different cases (just a few numbers in the table). It would be a good addition to the paper to add a few figures to get a better sense of how this glacier could change in the future, such as the spread of results, the spatial evolution of the ice front, or the

evolution of mass loss and discharge with time.

**3 Specific comments**

p.2 l.26: "with" → "leading to a"

p.2 l.26: "5.6 mm sea-level-rise" → "5.6 mm of sea level rise"

p.2 l.28: Why is the model unable to reproduce the winter calving? Is that a limitation of the model parameterization, the representation of calving (maybe the crevasse-depth calving is just one mode of calving and does not cover all the cases), the initial conditions? And what do you think are the consequences of the lack of winter calving?

p.2 Fig.1 caption: as mentioned above, using Jakobsson et al. (2012) is a rather unexpected choice, so it would be good to justify it and quantify the difference in bedrock elevation between this map and the other more standard maps.

p.2 l.30: the 17km/a speed is a seasonal speed happening over a few months in summer and not an annual velocity, it would be good to mention that.

p.3 l.44: "possessed" → "had"

p.3 l.57: "far faster" → "much faster"

p.4 l.64: "must be zero at the grounding line as it begins to float": I would rather say that it is zero under the floating tongue.

p.4 l.74: the melange does not really belong to the ice shelf or the glacier ("its floating melange"). I am also surprised to see "desintegration" associated to "melange" because the melange changes a lot seasonally, which is rather common, so I don't understand why these changes would be qualified of desintegration.

p.4 l.80: There is also a new study on Jakobshavn by Bondzio et al. (2018) using a

2d plan view model and a different calving parameterization, so it would be good to include these results in the comparison.

p.5 l.92: "BISICLES continuum ice sheet dynamics model" → "BISICLES ice sheet model"

p.5 l.100: "in hydrostatic equilibrium": the floating part only is in equilibrium"

p.6 l.107: "an approximate stress balance equation": replace by the name of the approximation and a reference as they are many difference approximations of the stress balance equations

p.8 l.136-137: How do you use the ocean conditions outside of the fjord to constrain the conditions inside the fjord and close to the grounding line?

p.8 l.140: "working hypothesis": there is not much in the discussion addressing this hypothesis and whether it was a valid one.

p.8 l.148: calving has been shown to be the main driver of the velocity Bondzio et al. (2017), so that changes in calving front positions could explain most of the dynamic changes of Jakobshavn Isbrae over the past three decades, so that changes in basal conditions indeed have a small impact for this glacier.

p.8 l.149-150: This is an interesting way to change the buttressing at the front. How different would the results be with another method to account for this buttressing?

p.8 l.152: How much buttressing does this represent? What would be the equivalent ice thickness needed to get a similar buttressing?

p.9 l.156-157: What is this runoff symbol?

p.9 l.163 and l.175: How do you link the far ocean field temperatures to the ocean temperature in the fjord and then the temperature at the grounding line?

p.10 l.178: The depth of the ocean temperatures used should depend on the geometry

of the fjord, including the highest depths of the sills. What is the depth of the ocean floor in Jakobshavn's fjord, and are there sills blocking the entry of the deepest waters?

p.10 l.186: "last 2 decades" while the rest of the paper rather show results since 2004.

p.10 l.187: "bedrock topography and ice thickness data in the year 2009 come from Gogineni et al. (2012)": As mentioned above, why use this dataset and not the more recent Bamber et al. (2013) or Morlighem et al. (2017) topography? Also, is this the same dataset as Jakobsson et al. (2012) shown of figure 1?

p.10 l.192-193: I am not sure to agree with this statement: the glacier was continuing to change following its ice tongue collapse as shown in Joughin et al. (2012).

Fig.3: the stiffening factor inferred with inverse problems is often difficult to physically explain and it is here mostly equal to 1. How different would the results be if it was just assumed to be equal to 1 everywhere and the basal traction coefficient was adjusted accordingly?

p.11 l.199-200: Why are the velocity from 2010 and the geometry from 2009 used in the inverse problem why the simulation is initialized to reproduce 2004?

p.12 l.201: "friction coefficient" and stiffening factor

p.12 l.210: Why is the model run until the profile matched the 1998 profile? I thought the target date was 2004?

p.12 l.211: How is it changed exactly?

p.12 l.214: Are you trying to get a stable state (How do you define stable by the way? What are the variables considered?) or to match the 2004 front position? I am also a little surprised that you are mentioning a "stable state" as the glacier has been continuously since at least the 90's.

p.14 l.254: Why is considered to be the total calving? Is it the difference between the ice front positions in 2013 and 2004 or the sum of the annual ice front change position?

The later sounds more appropriate to ensure that the timing of retreat is appropriate.

p.16 Fig.5a: There was an earlier mention of the relative stability after 2004, but this figure actually tends to show that there is not much stability at this period, with the summer front position retreating more every year.

p.18 Fig.6: Is it the bottom or the 300 m depth temperature?

p.19 l.312, 314, 318: What GCM was used to force RACMO?

p.19 l.303: change reference: Joughin et al. (2010) probably did not guess what would happen in the 2013-2017 period.

p.20 l.326-329: are these results shown on a figure?

p.20 Table 1 and lines 335-341: the numbers in the table and in the paragraph are somehow different (2068 vs 2029 Gt of mass loss by 2100 for example), but I might have missed something. It would also be appropriate to add the results from Bondzio et al. (2018) in the comparison here.

p.22 l.384: Does the fast flow go all the way to the sides of the fjord? It is a bit surprising to me that the bedrock provides so little resistance compared to the sides.

p.22 l.399: Is it possible to test this assumption of the impact of the shear-margin weakening mechanism?

p.26 l.448: There may also be some limitations associated with the initial conditions (ice too thick close to the ice front as shown on Fig.7).

p.26 l.462: "stimulate" → "simulate"

p.28 l.496: Did you use each GCM individually or used the mean of the 7 models?

**4 References**

Bamber, J. L., et al., A new bed elevation dataset for Greenland, Cryosphere, 7, 499–510, doi:10.5194/tc-7-499-2013, 2013.

Bondzio, J., M. Morlighem, H. Seroussi, T. Kleiner, M. Ruckamp, J. Mouginot, T. Moon, E. Larour, and A. Humbert, The mechanisms behind Jakobshavn Isbræ's acceleration and mass loss: A 3-D thermomechanical model study, Geophys. Res. Lett., 44, doi:10.1002/2017GL073309, 2017.

Bondzio, J. H., M. Morlighem, H. Seroussi, M. Wood, and J. Mouginot, Control of ocean temperature on Jakobshavn Isbræ's present and future mass loss, Geophys. Res. Lett., doi:10.1029/2018GL079827, 2018.

Jakobsson, M., et al., The International Bathymetric Chart of the Arctic Ocean (IBCAO) Version 3.0, Geophys. Res. Lett., 39, 1–6, doi:10.1029/2012GL052219, 2012.

Joughin, I., B. E. Smith, I. M. Howat, T. Scambos, and T. Moon, Greenland flow variability from ice-sheet-wide velocity mapping, J. Glaciol., 56, 416–430, 2010.

Joughin, I., B. E. Smith, I. M. Howat, D. Floricioiu, R. B. Alley, M. Truffer, and M. Fahnestock, Seasonal to decadal scale variations in the surface velocity of Jakob- shavn Isbrae, Greenland: Observation and model-based analysis, J. Geophys. Res., 117, 1–20, doi:10.1029/2011JF002110, 2012.

Morlighem, M., et al., BedMachine v3: Complete bed topography and ocean bathymetry mapping of Greenland from multi-beam echo sounding combined with mass conservation, Geophys. Res. Lett., 44(21), 11,051–11,061, doi:10.1002/2017GL074954, 2017GL074954, 2017.

---

## Author Comment (AC1) · 24 Apr 2019

In the reply, the referee's comments are in *italics*, our response is in normal text, quotes and modifications from the manuscript are in blue.

*Anonymous Referee #1*
*This paper presents a modelling study of Jakobshavn Isbrae using the ice flow model BISICLES. The model is initialised and calibrated to fit the observed front retreat and annual velocities between 2004 and 2013. Three parametrisations are used to control the position of the front:*
*• basal melting (Eq. 11),*
*• calving based on a crevasse depth criteria (Eq. 10),*
*• and a parametrisation meant to represent the buttressing of the ice melange in the front and that affect the driving stress (Eqs. 8-9) near the front.*
*Because it is a target for the calibration, the total observed front retreat is well reproduced by the model. The timing of the front variations with winter advance and summer retreat is captured by the model, however the seasonal variability is overestimated at the end of the period because the model does not reproduce the winter calving. Finally, the model is used to estimate the evolution during the 21st century. The results during the calibration are convincing and well discussed. However, I found that few points are missing for the description of the model and set-up, the initialization and calibration is relatively hard to follow as it involves many steps that have been implemented manually and the discussion mainly concentrates on what the model has and forget to include what is missing. I give more details below.*

*My main remark concerns the parametrisation of the buttressing by the ice melange. The boundary condition for the front is not mentioned in the model description but should be the difference between the force exerted by the ice and the back stress from the sea water.*

**Reply:** Sorry. We added boundary conditions into text:
Reflection boundary conditions were applied at the edge of each domain:
$$\boldsymbol{u} \cdot \boldsymbol{n} = 0, \qquad \boldsymbol{t} \cdot \nabla \boldsymbol{u} \cdot \boldsymbol{n} = 0, \qquad \nabla h \cdot \boldsymbol{n} = 0, \ (8)$$
where $\boldsymbol{n}$ is normal to a boundary and $\boldsymbol{t}$ is parallel to it. Normal stress across the calving front is equal to the hydrostatic water pressure there:
$$\boldsymbol{n} \cdot [\phi h \bar{\mu}(2\dot{\boldsymbol{\epsilon}} + 2\mathrm{tr}(\dot{\boldsymbol{\epsilon}})\mathbf{I})] - \boldsymbol{\tau}^b = \frac{1}{2}\rho_i \mathrm{g}\left(1 - \frac{\rho_i}{\rho_w}\right)h^2 \boldsymbol{n}. \ (9)$$

*It would seem natural to implement the effect of the ice melange as an additional back stress. The parametrisation implemented here modifies the driving stress near the front. More justification for this implementation is really required.*
**Reply:** Our parameterization of ice mélange buttressing is similar to Nick et al., (2013, Eq. S5) that alters the stress balance at calving front.

*Is this process really needed to reproduce the front variations? What appends if there is only the calving parametrisation?*
**Reply:** Mélange buttressing effects play a decisive role in the recent retreating and evolving of Jakobshavn (Joughin et al., 2004; Joughin et al., 2008; Vieli et al., 2011). Ice mélange buttressing does affect calving by altering the stress field that contributes to crevasse penetration depth (Eq. 12).

*Moreover, the effect is proportional to the fjord temperature and is thus continuously increasing in the future simulations. However, as the temperature increases, we may expect some kind of threshold where the ice melange disappears and its effects become negligible?*

**Reply:** Yes. There might be such a threshold. So far the physics of ice mélange is poorly understood, where no much clues can really help to speculate such a threshold.

*Few important informations are missing for the model description: What is the temperature field for the initialisation? Is the model thermo-mechanically coupled? What is the mesh resolution?*

**Reply:** Our model is not thermo-mechanically coupled with a fixed temperature field -10 °C. Our finest resolution is 500 m which cover the whole fast-flow-area (Fig. S1).

*The description of the initialisation is very hard to follow. For example for the step 2, we don't know what is the target to adjust β. In step 3, it is said that β from step 2 is used but that there is no calving, however β controls the calving criteria.*

**Reply:** Agreed, the statement was imprecise. The section 2.3 has been rewritten:

1) We solved the inverse problem for basal conditions (Eq. 7) and stiffening factor using 2010 velocities (Joughin et al., 2010) and 2009 geometry (Gogineni et al., 2012), following Cornford et al. (2015). Our friction coefficient and stiffening factor fields are shown in Fig. 3. Fig. S2 shows the discrepancy between observed velocity field (Joughin et al., 2010) and the velocity derived from the inversion.

2) Starting from the inversion of step 1, we let the model glacier evolve freely without calving and with zero SMB and with sub-shelf melting ($\gamma$=0.0238) forced by repeating the observed 2004 ocean temperature for 11 years until its surface elevation profile reached a state shown in Fig. S3.

3) We carried out several 10-year simulations each with different $\beta$ values estimated. These simulations were forced by repeatedly applying the 2004 seasonal climate forcing so that the glacier approaches a steady state. From these, we selected the $\beta$ that provided a calving front position closest to that observed in 2004. The best $\beta$ here is 0.034. This is our best guess for the 2004 state. The annual minimum extent of Jakobshavn retreats ~ 2 km from 2004 to 2005 following the loss of melange butressing, but then stabilizes until 2007 (Joughin et al. 2010). Annual maximum extents are stable over the 2004-2007 period. Front velocities increase slowly from 2004-2007 (~5.9% a$^{-1}$ Joughin et al. 2010), and the model simulated velocities increase by about 3% a$^{-1}$. This period of relative stability also makes 2004 a good time from which to start transient simulations.

Basal friction coefficient values downstream of the 2010 grounding line were set equal to that in the nearest 2010 grounded location. This was necessary because steps 2 and 3 involved grounding line advance beyond the region for which basal friction coefficients had been inferred. The geometry after this spin up procedure, and the friction coefficient and stiffening factor distribution from the inversion in step 1 were used as the initial condition for model calibration.

*It is said that $\gamma$ = 1, however in page 9, Eq. 11, it is said that $\gamma$ was derived from the 1985*

*observed submarine melt rate.*

**Reply:** *We rewrote this paragraph:*

The parameters in the model, α, β and γ representing mélange buttressing, crevasse depth sensitivity to surface runoff, and shelf melt sensitivity to ocean temperatures need to be estimated. The measured relationship between ocean temperatures and sub-shelf melt rate (Motyka et al., 2011) gives the value of γ to be 0.238. We tune parameters α and β manually to best reproduce Jakobshavn Isbræ's calving front position and surface velocity evolution for the 10 year period 2004-2013. Reproducing the total retreat distance and the temporary stable state after 2012 were secondary desirable features to match. The best set of parameters are $\alpha_1$=0.82, $\alpha_2$=0.111, β=0.0638. Since these values come from a manual search we do not claim them to be the best in all parameter space. We assess model sensitivity to the parameter values next.

*The discussion mainly focuses on the effect of the shear margins and the fact that due to the non linearity in the ice flow law the effective viscosity decreases as the strain-rates increase. This mechanism is described as a positive feedback, however I'm not sure that this is the right term, as both the velocities and the strain-rates are the results of the force balance equation, so that the velocities and the effective viscosity depends on the model parameters and boundary conditions. But is it difficult to describe this as a feedback as they both are solution of the same equation.*

**Reply:** Vertically varying effective viscosity $\mu$ is solved by Eq. 6, instead of the stress balance equation (Eq. 3).

*However, the results are certainly dependent of the ice flow law and the value of the stress exponent, and we certainly may expect that the results would be different with a linear flow relation, or, as discussed, a flow band model that would parametrised the lateral drag.*

**Reply:** Yes. Our chosen stress exponent is quite common.

*The mesh resolution might also influence the results as the resolution should be sufficient to properly capture the steep velocity gradients to represent this effect.*

**Reply:** Our finest resolution of 500 m well covers the main trunk and shear margins (Fig. S1).

*In addition, the comparison with Bondzio et al. (2017) might be a bit confusing as Bondzio et al. include the thermo-mechanical coupling (which is absent here?), and they report that the warming from shear heating accounts for 20 to 30% of the decrease in effective viscosity. There is also several mechanisms that could affect the viscosity of the shear margins with potential feedbacks, this includes damage, cryo-hydrologic warming, anisotropy etc... This could be discussed also.*

**Reply:** Our model is not thermo-mechanical coupled. The periods for viscosity drop calculations are not quite overlapped. Here we only attempt to roughly verify our results by cross model comparison. We add more discussion in section 4:

Several absent processes in our model could affect ice viscosity. Crevasses saturated by surface melt water within the shear margins of Jakobshavn occurred and last in summer that is visible on satellite images (Lampkin et al., 2013). These melt water can transfer heat throughout the

ice column through discharge within crevasses and moulins thus soften the ice (Phillips et al., 2010). Consideration a continuum damage model in BISICLES would further exaggerate the shear margin weakening as it rise the non-linear dependence of strain rates to stress fields (Sun et al., 2017).

*Finally it would be also interesting to see or at least discuss how the model results are sensitive to other uncertainties in the model; this includes the description of the bedrock and the basal friction law, especially the linear assumption that is used here.*

**Reply:** Lines added in Model improvements:

Ice thickness and basal topography with resolution of 150 m became available for main outlet glaciers of Greenland (Morlighem et al., 2017) recently. This allows finer mesh resolution in modeling works which would be expected to reveal more details of ice-stream especially on perpendicular-to-flow direction, including more precise shear-margin-weakening and single calving near side walls. Other than our simple assumption of basal drag (Eq. 7), implementing a physics-based basal sliding law (Schoof, 2010; Gagliardini et al., 2014; Tsai et al., 2015) would produce more speedup and retreats in model results as dynamic thinning constantly reduce the effective pressure which leads to lower basal shear stress.

---

## Author Response (AR1)

In the reply, the referee's comments are in *italics*, our response is in normal text, quotes and modifications from the manuscript are in blue.

***Anonymous Referee #1***
*This paper presents a modelling study of Jakobshavn Isbrae using the ice flow model BISICLES. The model is initialised and calibrated to fit the observed front retreat and annual velocities between 2004 and 2013. Three parametrisations are used to control the position of the front:*
* *basal melting (Eq. 11),*
* *calving based on a crevasse depth criteria (Eq. 10),*
* *and a parametrisation meant to represent the buttressing of the ice melange in the front and that affect the driving stress (Eqs. 8-9) near the front.*
*Because it is a target for the calibration, the total observed front retreat is well reproduced by the model. The timing of the front variations with winter advance and summer retreat is captured by the model, however the seasonal variability is overestimated at the end of the period because the model does not reproduce the winter calving. Finally, the model is used to estimate the evolution during the 21st century. The results during the calibration are convincing and well discussed. However, I found that few points are missing for the description of the model and set-up, the initialization and calibration is relatively hard to follow as it involves many steps that have been implemented manually and the discussion mainly concentrates on what the model has and forget to include what is missing. I give more details below.*

*My main remark concerns the parametrisation of the buttressing by the ice melange. The boundary condition for the front is not mentioned in the model description but should be the difference between the force exerted by the ice and the back stress from the sea water.*
**Reply:** Yes you are right. We added boundary conditions into text:
Reflection boundary conditions were applied at the edge of each domain:
$$\boldsymbol{u} \cdot \boldsymbol{n} = 0, \qquad \boldsymbol{t} \cdot \nabla \boldsymbol{u} \cdot \boldsymbol{n} = 0, \qquad \nabla h \cdot \boldsymbol{n} = 0, \quad (8)$$
where $\boldsymbol{n}$ is normal to a boundary and $\boldsymbol{t}$ is parallel to it. Normal stress across the calving front is equal to the hydrostatic water pressure there:
$$\boldsymbol{n} \cdot [\phi h \bar{\mu}(2\dot{\epsilon} + 2\mathrm{tr}(\dot{\epsilon})\mathbf{I})] - \boldsymbol{\tau}^b = \frac{1}{2}\rho_i \mathrm{g}\left(1 - \frac{\rho_i}{\rho_w}\right)h^2\boldsymbol{n}. \quad (9)$$

*It would seem natural to implement the effect of the ice melange as an additional back stress. The parametrisation implemented here modifies the driving stress near the front. More justification for this implementation is really required.*
**Reply:** Yes that's one way, but our parameterization of ice mélange buttressing is similar to Nick et al., (2013, Eq. S5) which also alters the stress balance at calving front. So our method is established in the literature. We cite Nick in this part.

*Is this process really needed to reproduce the front variations? What appends if there is only the calving parametrisation?*

**Reply:** Mélange buttressing effects plays a decisive role in the recent retreating and evolving of Jakobshavn (Joughin et al., 2004; Joughin et al., 2008; Vieli et al., 2011). Ice mélange buttressing does affect calving by altering the stress field that contributes to crevasse penetration depth (Eq. 12). Our buttressing parameterization gives a longitudinal resistance that is 18% of the driving force at the calving front (Eq. 10), for the instance of 2004.

*Moreover, the effect is proportional to the fjord temperature and is thus continuously increasing in the future simulations. However, as the temperature increases, we may expect some kind of threshold where the ice melange disappears and its effects become negligible?*

**Reply:** The presence of mélange may scale with temperature to some extent, but it is unlikely to exhibit threshold behavior, because ocean water cannot instantaneously melt ice. Also, increased calving fluxes will oppose ocean warming. These processes are not yet well enough understood for their impact on mélange buttressing to be quantified.

*Few important informations are missing for the model description: What is the temperature field for the initialisation? Is the model thermo-mechanically coupled? What is the mesh resolution?*

**Reply:** Our model is not thermo-mechanically coupled with a fixed temperature field - 10 °C. Our finest resolution is 500 m which cover the whole fast-flow-area, and we show an example in the new Fig. 1C.

[Figure]

**Figure 1. A) Greenland ice sheet flow speeds from Joughin et al. (2018), with the Jakobshavn drainage basin outlined by the solid black line and the area shown in panel B by the dashed box. B) Ilulissat Fjord and Disko Bay bathymetry from Jakobsson et al. (2012), with the CTD (Conductivity Temperature Depth) site used for ocean temperature here marked by the red star. C) Example of the mesh used with finest resolution of 500 m with modeled velocities at the beginning of 2004.**

*The description of the initialisation is very hard to follow. For example for the step 2, we don't know what is the target to adjust β. In step 3, it is said that β from step 2 is used but that there is no calving, however β controls the calving criteria.*

**Reply:** Agreed, the statement was imprecise. The section 2.3 has been rewritten:

1) We solved the inverse problem for basal conditions (Eq. 7) and stiffening factor using 2010 velocities (Joughin et al., 2010) and 2009 geometry (Gogineni et al., 2012), following Cornford et al. (2015). Our friction coefficient and stiffening factor fields are shown in Fig. 3. Fig. S1 shows the discrepancy between observed velocity field (Joughin et al., 2010) and the velocity derived from the inversion.

2) Starting from the inversion of step 1, we let the model glacier evolve freely without calving and with zero SMB and with sub-shelf melting ($\gamma$=0.0238) forced by repeating the observed 2004 ocean temperature for 11 years until its surface elevation profile reached a state shown in Fig. S2.

3) We carried out several 10-year simulations each with different $\beta$ values estimated. These simulations were forced by repeatedly applying the 2004 seasonal climate forcing so that the glacier approaches a steady state. From these, we selected the $\beta$ that provided a calving front position closest to that observed in 2004. The best $\beta$ here is 0.034. This is our best guess for the 2004 state. The annual minimum extent of Jakobshavn retreats ~ 2 km from 2004 to 2005 following the loss of melange butressing, but then stabilizes until 2007 (Joughin et al. 2010). Annual maximum extents are stable over the 2004-2007 period. Front velocities increase slowly from 2004-2007 (~5.9% $a^{-1}$ Joughin et al. 2010), and the model simulated velocities increase by about 3% $a^{-1}$. This period of relative stability also makes 2004 a good time from which to start transient simulations.

Basal friction coefficient values downstream of the 2010 grounding line were set equal to that in the nearest 2010 grounded location. This was necessary because steps 2 and 3 involved grounding line advance beyond the region for which basal friction coefficients had been inferred. The geometry after this spin up procedure, and the friction coefficient and stiffening factor distribution from the inversion in step 1 were used as the initial condition for model calibration.

[Figure]

Figure S1. A) Velocity discrepancy (velocity from inversion - observed) and B) the observed velocity field (Joughin et al., 2010).

[Figure]

**Figure S2. Profiles of surface elevation during the initialization procedure (section 2.3) step 3. Black solid line and black dashed line show the known profiles taken in the 1990s (Bamber et al., 2001) and 2010 (Gogineni et al., 2012) respectively. The profile with legend '1st yr' is the final state of section 2.3 step 2. The profile '7th yr' is the geometry rebuilt for 2004's Jakobshavn, which is the initial state for later simulations.**

*It is said that γ = 1, however in page 9, Eq. 11, it is said that γ was derived from the 1985 observed submarine melt rate.*

**Reply:** We were unclear and so rewrote this paragraph:

The parameters in the model, α, β and γ representing mélange buttressing, crevasse depth sensitivity to surface runoff, and shelf melt sensitivity to ocean temperatures need to be estimated. The measured relationship between ocean temperatures and sub-shelf melt rate (Motyka et al., 2011) gives the value of γ to be 0.238. We tune parameters α and β manually to best reproduce Jakobshavn Isbræ's calving front position and surface velocity evolution for the 10 year period 2004-2013. Reproducing the total retreat distance and the temporary stable state after 2012 were secondary desirable features to match. The best set of parameters are $\alpha_1$=0.82, $\alpha_2$=0.111, β=0.0638. Since these values come from a manual search we do not claim them to be the best in all parameter space. We assess model sensitivity to the parameter values next.

*The discussion mainly focuses on the effect of the shear margins and the fact that due to the non linearity in the ice flow law the effective viscosity decreases as the strain-rates increase. This mechanism is described as a positive feedback, however I'm not sure that this is the right term, as both the velocities and the strain-rates are the results of the force balance equation, so that the velocities and the effective viscosity depends on the model parameters and boundary conditions. But is it difficult to describe this as a feedback as they both are solution of the same equation.*

**Reply:** OK, the mechanism is due to the non-linear rheology of the ice, so we change the text to "This mechanism is due to the non-linear rheology of the ice in the fast flow region".

*However, the results are certainly dependent of the ice flow law and the value of the stress exponent, and we certainly may expect that the results would be different with a linear flow relation, or, as discussed, a flow band model that would parametrised the lateral drag.*
**Reply:** Agreed and addressed by the change above.

*The mesh resolution might also influence the results as the resolution should be sufficient to properly capture the steep velocity gradients to represent this effect.*
**Reply:** Yes, our finest resolution is 500 m and covers the main trunk and shear margins (Fig. 1C).

*In addition, the comparison with Bondzio et al. (2017) might be a bit confusing as Bondzio et al. include the thermo-mechanical coupling (which is absent here?), and they report that the warming from shear heating accounts for 20 to 30% of the decrease in effective viscosity. There is also several mechanisms that could affect the viscosity of the shear margins with potential feedbacks, this includes damage, cryo-hydrologic warming, anisotropy etc... This could be discussed also.*
**Reply:** Agreed. Our model is not thermo-mechanical coupled. The periods used by Bondzio are not are not quite the same as ours. Here we only attempt to roughly verify our results by cross model comparison. We add more discussion in section 4:
Several absent processes in our model could affect ice viscosity. Crevasses saturated by surface melt water within the shear margins of Jakobshavn are visible on satellite images (Lampkin et al., 2013). This melt water can transfer heat throughout the ice column through discharge within crevasses and moulins thus soften the ice (Phillips et al., 2010). Incorporating a continuum damage model in BISICLES would further exaggerate the shear margin weakening as it raises the non-linear dependence of strain rates to stress fields (Sun et al., 2017).

*Finally it would be also interesting to see or at least discuss how the model results are sensitive to other uncertainties in the model; this includes the description of the bedrock and the basal friction law, especially the linear assumption that is used here.*
**Reply:** We added these lines in Model improvements to cover this issue:
Ice thickness and basal topography with resolution of 150 m became available for main outlet glaciers of Greenland (Morlighem et al., 2017) recently (Fig. S3). This eases finer mesh resolution to be used for modeling which then might reveal more details of ice-stream behavior especially perpendicular-to-flow direction, including more precise shear-margin-weakening and calving near side walls. Our assumption of simple Weertman basal drag (Eq. 7) may be improved by implementing a physics-based basal sliding law (Schoof, 2010; Gagliardini et al., 2014; Tsai et al., 2015), although basal drag accounts for only about 2% of present-day buttressing (Shapero et al., 2016). An improved sliding relation would likely produce more speedup and retreats in model results as dynamic thinning constantly reduce the effective pressure which leads to lower basal shear stress.

[Figure]

**Figure S3. Bed elevation from BedMachine v3 (Morlighem et al., 2017) minus those from (Gogineni, 2012) used in this paper.**

*Referee #2*

**General Comments**: *The model approach resembles previous studies by Muresan et al. 2016 by using ocean temperatures as a forcing to a dynamic ice flow model. However, Xiaoran Guo et al. expands the approach by going into more detail on seasonality and viscosity changes, while also starting their model in 2004 (not in 1990 as Muresan et al does) where they provide evidence that there is a shift in flow regime. Thus, there is a scientific advance within the field, by exploring ways to improve methods for modelling the behaviour of fast flowing ice streams. These types of model studies requires a lot of technical settings and tuning of the model which is very complicated and hard to explain in an easy-to-understand way. However, in order to satisfy the demand of traceability of results, this is the most important part of the paper. The model setup sections are not doing this sufficiently, in their current state.*

**Specific comments**: *Model description sections: Initialisation and calibration should be improved to make it clearer exactly what has been done. In particular I am missing information about what basal and surface geometry is used in the inversion process and also how values for basal friction and ice softness are derived. Furthermore, I am curious about the mesh resolution used in the model and in particular how this looks across the shear margins.*

**Reply:** Yes this is something the other referees mentioned as well, so we have extensively rewritten section 2.3.

First we added the mesh resolution and refer to a new Fig.1C from line 99:

… but the fast flow area is only around 10 km in width. We use a highest mesh resolution of 500 m that covers the whole fast-flow-area including the shear margin (Fig. 1C), while the rest of the glacier has 1000 m resolution.

[Figure]

**Figure 2. A) Greenland ice sheet flow speeds from Joughin et al. (2018), with the Jakobshavn drainage basin outlined by the solid black line and the area shown in panel B by the dashed box. B) Ilulissat Fjord and Disko Bay bathymetry from Jakobsson et al. (2012), with the CTD (Conductivity Temperature Depth) site used for ocean temperature here marked by the red star. C) Example of the mesh used with finest resolution of 500 m with modeled velocities at the beginning of 2004.**

We change the description of the initialization procedure (line 199 onwards), to answer questions on geometry and method used for inversion.

1) We solved the inverse problem for basal conditions (Eq. 7) and stiffening factor using 2010 velocities (Joughin et al., 2010) and 2009 geometry (Gogineni et al., 2012), following Cornford et al. (2015). Our friction coefficient field is shown in Figure 3. Figure S2 shows the discrepancy between observed velocity field (Joughin et al., 2010) and the velocity derived from the inversion.

**Basal geometry**: *It is not stated anywhere what basal geometry is used. As, the authors also state in the discussion, geometry is the most important factor for ice stream*

*stability and thus the results of the retreating calving front should furthermore be mapped on top of a basal geometry map in 2d plan view (seen from above). The retreat pattern relation to basal geometry should be discussed in relation to other studies modelling the future behaviour of Jakobshavns Isbræ.*

**Reply:** Our basal topography data come from (Gogineni et al., 2012). We solved the inverse problem for basal conditions (Eq. 7) and stiffening factor using 2010 velocities (Joughin et al., 2010) and 2009 geometry (Gogineni et al., 2012).

We plotted Jakobshavn's retreat in Fig. 5 in 1-d considering the convenience of comparison with previous studies (Nick et al., 2013; Muresan et al., 2016). We added a panel to Fig. 7 showing modelled front retreats along its basal trough using the best set of parameters, as requested by the referee.

[Figure]

**Figure 7. Modeled profiles of (A) January velocity and (B) January surface elevation along the center-flow-line (purple dash line in panel C) of Jakobshavn Isbræ from 2004 to 2099 for the RCP4.5 scenario. Bedrock elevation is shown in black. Black dotted line is the surface elevation profile extracted from radar data measured around 2010 (Gogineni et al., 2012). Profiles are shown at intervals of 1 years. Profiles are color-coded in the legend and range from blue to green and red. (C) Modeled July front positions (color bar) over its bedrock**

(grayscale bar) at intervals of 2 years.

***In relation to starting in 2004****: To my understanding, and also what you describe for the model, a stiff ice mélange has a butressing effect. Thus, it seems strange to me that the glacier is stable from 2004 and onwards, if it just lost an important butressing?*
**Reply:** Yes, our description was confusing and imprecise. The annual minimum extent of Jakobshavn retreats ~ 2 km from 2004 to 2005, but then stabilizes until 2007 (Joughin et al. 2010). Front velocities increase slowly from 2004-2007 (~5.9% a$^{-1}$). Annual maximum extents are stable over the 2004-2007 period. We change the text as :
The annual minimum extent of Jakobshavn retreats ~ 2 km from 2004 to 2005 following the loss of melange butressing, but then stabilizes until 2007 (Joughin et al. 2010). Front velocities increase slowly from 2004-2007 (~5.9% a$^{-1}$). Annual maximum extents are stable over the 2004-2007 period. This also makes 2004 a good time from which to start transient simulations.

***Line by line comments****: Section 1 Generally, there is confusion about the definition of a floating ice shelf and a stiff ice mélange throughout the section.*
**Reply:** We cannot understand the confusion the referee mentions. We are using standard definitions: ice mélange is the broken bergs and sea ice in front of the calving front. Ice tongue and ice shelf mean the floating glacier ice that is mechanically coupled with the inland glacier.

***Line 70-72****: Needs a reference*
**Reply:** Done:
However, in the Jakobshavn case, both Weertman and Coulomb sliding produce very similar fluxes because the basal shear stresses along the main trough are typically only 2 % of the driving force (**Shapero et al., 2016**).

***Section 2 Line 102****: what basal map do you use? Line 123-124: Please refer us to a study where the method of solving the inverse problem where two unknown is discussed (or explain in detail here how that would work, and how you can trust the outcome). I think this is an important point as viscosity is non-linear.*
**Reply:** The method is well established in BISICLES, and described e.g. by Cornford et al. (2015). It is possible because vertically integrated shear is used rather than a full Stokes formulation as in e.g. ELMER/ice. We solved the inverse problem for basal conditions (Eq. 7) and stiffening factor using 2010 velocities (Joughin et al., 2010) and 2009 geometry (Gogineni et al., 2012), following Cornford et al. (2015).

***Section 2.2****: Should just be titled Forcing (and not climate forcing*
**Reply:** Done. We change the title to 'Forcing'.

*Line 135: What is CTD?*
**Reply:** CTD is defined in Fig. 1 caption as "**CTD (Conductivity Temperature Depth)**".

*Line 136: At what depth is the ocean temperature a good approximation?*
**Reply:** We use 300 m depths. Cowton et al. (2018) achieved success in simulating the terminus position and yearly variability of 10 glaciers along the east coast of Greenland using mean 200-400 m depth temperatures from reanalysis data.
We use ocean temperatures at depth ~ 300 m collected from a CTD site close to the mouth of Ilulissat fjord (Fig.1) as an approximation of ocean temperatures near the glacier grounding line.

*Line 153: Use alpha1 and alpha2 instead of the calibrated numbers*
**Reply:** Equation 9 now reads: $\alpha = \alpha_1 + \alpha_2 T.$ (9)

*Line 157-160: MAR is used to estimate the runoff in equation 10. Later on Racmo is used as forcing. It is not clear why you use two different models, and when they are used.*
**Reply:** Yes we clarify this after line 183 at the end of section 2.2. For the period 2004-2014, SMB and surface water run-off forcing come from MAR model outputs. Because RACMO outputs cover only the period of 2006-2099. For the period 2015-2099, our SMB and run-off forcing are from RACMO outputs. We use the overlapping period 2006-2014 to correct the bias between two models outputs.

*Line 169: Make it clear that it is your model your are talking about*
**Reply:** Here are talking about the RACMO forcing as said as the first word on line 170. Hence not from our simulation.

*Line 169: Write out SMB*
**Reply:** SMB is already defined in Line 111: where $M_s$, $M_b$ are surface mass balance (SMB)…

*Line 174-177: Please state in what equation this ocean forcing goes into.*
**Reply:** Ocean temperature forcing affects mélange buttressing and sub-shelf melting (Eq. 10, 13). Ocean forcing in Equations 10 and 13 should relate to temperatures off the continental shelf close to the fjord mouth.

**Section 2.3**
*Line 187: The dataset described here is only 2d, your model is in 3d, so I am not sure what you are using this for?*
**Reply:** The sentence is: "Detailed bedrock topography and ice thickness data in the year 2009 come from Gogineni et al. (2012)." Bedrock topography is a 2d dataset, ice thickness is the vertical direction dataset, we use these in our 3d model. BISICLES calculates surface elevations by Eq.1.

*Line 188: to my understanding, the sudden disappearance*
**Reply:** We prefer disintegration as it cannot have simply disappeared.

*Line 202: please remind us what beta is*
**Reply:** This section has been completely rewritten see reply to line 218 below.

*Line 209-210: What does similar mean? How far off are we talking here? And please state why you use the 1998 profile when the model is starting in 2004.*
**Reply:** New Fig. S2 (see answer to line 218) shows the surface elevation profiles. We use that because the geometry in 2004 is unknown. In fact we only need the height at the grounding line, not the whole profile.

*Line 214: Why is it the 8th, needs clarification.*
**Reply:** Agreed and rephrased (line 218).

*Line 217: Aha, good to know already in line 209-210*
**Reply:** Agreed and rephrased (line 218).

*Line 218-219: The glacier is definitely not in steady state in 2004, please rephrase 2.4 Model calibration. This section is very confusing to me. I think it needs a rewrite to become clearer.*
**Reply:** Agreed. We rewrote the section 2.3:
1) We solved the inverse problem for basal conditions (Eq. 7) and stiffening factor using 2010 velocities (Joughin et al., 2010) and 2009 geometry (Gogineni et al., 2012), following Cornford et al. (2015). Our friction coefficient and stiffening factor fields are shown in Fig. 3. Fig. S1 shows the discrepancy between observed velocity field (Joughin et al., 2010) and the velocity derived from the inversion.
2) Starting from the inversion of step 1, we let the model glacier evolve freely without calving and with zero SMB and with sub-shelf melting ($\gamma$=0.0238) forced by repeating the observed 2004 ocean temperature for 11 years until its surface elevation profile reached a state shown in Fig. S2.
3) We carried out several 10-year simulations each with different $\beta$ values estimated. These simulations were forced by repeatedly applying the 2004 seasonal climate forcing so that the glacier approaches a steady state. From these, we selected the $\beta$ that provided a calving front position closest to that observed in 2004. The best $\beta$ here is 0.034. This is our best guess for the 2004 state. The annual minimum extent of Jakobshavn retreats ~ 2 km from 2004 to 2005 following the loss of melange butressing, but then stabilizes until 2007 (Joughin et al. 2010). Annual maximum extents are stable over the 2004-2007 period. Front velocities increase slowly from 2004-2007 (~5.9% $a^{-1}$ Joughin et al. 2010), and the model simulated velocities increase by about 3% $a^{-1}$. This period of relative stability also makes 2004 a good time from which to start transient simulations.

Basal friction coefficient values downstream of the 2010 grounding line were set equal to that in the nearest 2010 grounded location. This was necessary because steps 2 and 3 involved grounding line advance beyond the region for which basal friction coefficients had been inferred. The geometry after this spin up procedure, and the friction coefficient and stiffening factor distribution from the inversion in step 1 were used as the initial condition for model calibration.

[Figure]

**Figure S1. A) Velocity discrepancy (velocity from inversion - observed) and B) the observed velocity field (Joughin et al., 2010).**

[Figure]

**Figure S2. Profiles of surface elevation during the initialization procedure (section 2.3) step 3. Black solid line and black dashed line show the known profiles taken in the 1990s (Bamber et al., 2001) and 2010 (Gogineni et al., 2012) respectively. The profile with legend '1st yr' is the final state of section 2.3 step 2. The profile '7th yr' is the geometry rebuilt for 2004's Jakobshavn, which is the initial state for later simulations.**

*Line 235: rephrase sentence Line 235-245: I am confused about this whole paragraph. The following paragraph (Line 246-259) is better structured, could this perhaps be the start of the section?*

**Reply:** We prefer to discuss optimal parameter values and then sensitivities, so we rewrite the paragraph:

The parameters in the model, α, β and γ representing mélange buttressing, crevasse depth sensitivity to surface runoff, and shelf melt sensitivity to ocean temperatures need to be estimated. The measured relationship between ocean temperatures and sub-shelf melt rate (Motyka et al., 2011) gives the value of γ to be 0.238. We tune parameters α and β manually to best reproduce Jakobshavn Isbræ's calving front position and surface velocity evolution for the 10 year period 2004-2013. Reproducing the total retreat distance and the temporary stable state after 2012 were secondary desirable features to match. The best set of parameters are $\alpha_1=0.82$, $\alpha_2=0.111$, $\beta=0.0638$. Since these values come from a manual search we do not claim them to be the best in all parameter space. We assess model sensitivity to the parameter values next.

*Line 274-284*: *This whole paragraph needs clarification*
**Reply:** This paragraph (Line 274-284) has been rewritten:
The two biggest mismatches occur with the 2007 and especially 2013 velocities (Fig. 5). 2013 has the lowest simulated surface water run-off (Fig. 2) of all the years since 2004. The Benn calving model we use is sensitive to runoff, with reduced run-off leading to lower crevasse-penetration-depth and reduced terminus fracturing thus increasing its buttressing force. Furthermore 2013 had relatively cool ocean temperatures which were lower than the average of 2004-2013. The cool ocean temperatures also increased buttressing, leading to low simulated annual mean velocities. Jakobshavn Isbræ did not in fact slow down very much in 2013 because there were calving events that are unrepresented in our model. The relevant mechanisms are discussed later.

*Figure 5A*: *How is the calving front retreat defined? Is it just a comparison at points along a centre flow line? And it this representative of the general retreat?*
**Reply**: Yes, it is defined as the distance along the center-flow-line. See Fig. 7 caption.

*Section 3 Figure 7*: *I would be more interested in seeing the retreat from above, the center flowline bedmap does not explain the stop of retreat.*
**Reply**: Agreed. We add a new panel to Fig. 7 showing the 2-d map and new figs 8 and 9 showing more details that help explain the retreat cessation.
Examination of the change in velocities during the simulation (Fig. 9) suggests that the explanation for this stability is strong flow convergence near the future glacier front that largely offsets dynamic thinning. Notice that the South side of the fast-flow-area in 20[th] century was quite close to ice-free land, while in later half of this century convergent flow in the South is fed by a substantial area of ice stream.

[Figure]

**Figure 7. Modeled profiles of (A) January velocity and (B) January surface elevation along the center-flow-line (purple dash line in panel C) of Jakobshavn Isbræ from 2004 to 2099 for the RCP4.5 scenario. Bedrock elevation is shown in black. Black dotted line is the surface elevation profile extracted from radar data measured around 2010 (Gogineni et al., 2012). Profiles are shown at intervals of 1 years. Profiles are color-coded in the legend and range from blue to green and red. (C) Modeled July front positions (color bar) over its bedrock (grayscale bar) at intervals of 2 years.**

[Figure]

**Figure 9. Simulated velocity vectors in 2004 (pink vectors) with their magnitudes (right color bar) and velocity difference between 2004 and 2099 (2099's minus 2004's, black vectors), for clarity vector lengths are clipped at 5 km a⁻¹.**

***Line 322****: Make it clear that you are talking about you model version of Jakobshavn Isbræ.*
**Reply**: OK: In our modelled results under this forcing, Jakobshavn Isbræ continues its retreat (Fig. 7) for 18 years after 2013, producing a total grounding line retreat of ~18 km upstream.

***Section 4:*** *It confused me a bit that you called this Discussion as a lot of important results are also presented here.*
**Reply**: Our modelled seasonal cycle of shear margin weakening is the main distinguishable feature comparing with previous studies. So it is convenient to put results, such as viscosity changes, here to make comparison with others.

***Line 370-372****: This sentence does not make sense to me, does freshly calved ice bergs really provide any resistance?*
**Reply**: Agreed. We deleted this sentence.

***Line: 376-377****: here you call it a stiff ice mélange, I think you should use this term throughout, especially in the intro.*

**Reply**: We are using standard definitions: ice mélange is the broken bergs and sea ice in front of the calving front. Ice tongue and ice shelf mean the floating glacier ice that is mechanically coupled with the inland glacier.

*Section 4.2 I am a bit confused, are the results of changes in the effective viscosity shown in figure 8 results from your forward run? And if so, how does the fact that you are keeping ice softness constant influence these results? I think there must be an effect in the softening from the thermodynamics as well?*

**Reply**: Yes. Fig. 10 (old fig. 8) shows results from the forward run. We plotted $\Phi \mu$ (Eq. 5) in Fig. 10. Our Ice stiffness factor $\Phi$ is fixed but ice viscosity $\mu$ varies in time. Bondzio et al., 2017 used a thermomechanical ice flow model to evolve the ice viscosity, which depends on a damage parameter that soften the ice in the shear margins. But their damage parameter also stays constant in time. Thus both our and their models only consider the contribution from strain rate weakening in time to evolving viscosity. Thermodynamics could play some role in changing viscosity, presumably if the ice temperatures increased over time. We suppose that this would be a minor effect compared with mechanical softening, and the temperature of the ice is fixed in our model.

*Section 4.3 Good to have comparisons with previous results, I think a key point, which you focus very little on, is that the retreat stops in the same area in all the studies (if I understood this correctly)? I think that if you also add figures showing basal geometry and retreat as suggested earlier, this point is easily added.*

**Reply:** In previous studies, Nick et al. (2013) and Muresan et al. (2016) didn't reproduce Jakobshavn's retreats to the bottom of its over-deepened basin, which we did (Fig. 5). We added the 2-d views of the past and predicted front retreats to Fig. 7C and Fig. 8. In our range of future projections we find retreat is slowed in the same places and new Fig. 9 tries to explain the reason for the retreat stoppage.

[Figure]

**Figure 8. Upper and lower estimates of July front positions within this century with colors indicating the date (color bar) for A) lower bound with scalings of (1,0.8) and the HadGEM-ES forcing B) upper bound of mass loss projection with (α, γ) parameter scalings of (1.2,1), and the 7-model ensemble climate forcing.**

*Line 498-499: What do you mean by two-dimensional ice flow patterns?*
**Reply:** Here 'ice flow patterns' refers to 'ice velocity and viscosity structures'. Thus:
We successfully model two-dimensional ice velocity and viscosity structures and their seasonal variations for Jakobshavn Isbræ, which are missing from several previous modeling studies. Moreover, capturing these two-dimensional structures allows us to handle the influence of horizontal velocity shear on effective ice viscosity, which impacts on speedup processes of Jakobshavn Isbræ.

*1 Summary statement*

*The manuscript "Simulated retreat of Jakobshavn Isbrae during the 21st century" by X. Guo and colleagues presents results on the simulation of Jakobshavn Isbrae over the 21st century, calibrated to match its current configuration and recent evolution. The model includes buttressing provided by the ice melange and a calving law based on crevasse-depth, and the forcings are based on Global Climate Models (GCMs). The results suggest that the glacier will continue to retreat and lose mass during the 21st century, reaching 5.6 mm of sea level equivalent by the end of the century.*

*The paper is well written, usually easy to follow (except for the initialization procedure that is quite complicated), and the figures appropriate. However, some additional explanations are needed to understand the choices made for the calibration of several parameters, for some of the datasets used, or for the initialization procedure. Furthermore, only a couple of figures show the evolution of the glacier over a flow line for one given simulation of the ensemble. It would be valuable to show the spread of the model results for the different parameters used and the different forcings, but also to show the spatial evolution of the ice front not just on a flow line but for the entire basin. Finally, the authors mention that the calving law based on crevasse-depth prevents the calving of the glacier once the thickness becomes too large. This is the contrary of what is physically expected: a tall cliff with a large height above sea level leads to more calving, so there is no reason for the calving to get reduced towards the end of the simulations.*

*2 Major comments*

*The bedrock and bathymetry used come from Jakobsson et al. (2012) and Gogineni et al. (2012), while the newer bedrock elevation maps of Greenland typically used in ice sheet modeling are Bamber et al. (2013) and Morlighem et al. (2017), so it is a rather interesting choice. There are probably good reasons for using these maps, but they are not well explained. It would be good quantify the impact of this choice on the simulations compared to other choices, or at least explain the differences expected.*

**Reply:** Our glacier geometry data (Gogineni et al., 2012) is derived from the same institution's products as used in BedMachine V3 with data processed by the Center for Remote Sensing of Ice Sheets (CReSIS, Leuschenetal., 2010 updated 2016). For computing resource considerations, we chose the earlier product because it has 500-meter-resolution. The difference between these two bedrock elevations are shown in Fig. S3.

[Figure]

**Figure S3. A) Bed elevation from BedMachine v3 (Morlighem et al., 2017) minus those from (Gogineni, 2012) used in this paper.**

*The calving law is based on crevasse-depth based calving only, so that calving happens when deep surface crevasses develop in the presence of surface water. Is this representation of calving sufficient to represent the different types of calving throughout the year and as the glacier retreats to deeper grounds? It seems that the model is not able to simulate calving in winter that is becoming important towards the end of the simulation. Would a different parameterization of calving lead to different results? This is a rather important question as it contradicts the marine ice cliff instability that predicts faster and faster retreat as glaciers retreat to deeper grounds and the ice thickness increases. So what is the impact of choosing a crevasse-depth based calving? This should be addressed in more details in the discussion.*

**Reply:** As we state in our paper at quite some length, our model clearly does not represent winter calving, which does become more important later in observational record. This is beyond dispute.

Winter calving is poorly understood. Its mechanism could be non-hydrostatic processes including the terminus uplifting due to super-buoyant condition with the opening of basal crevasses (James et al, 2014; Xie et al., 2016; Benn et al., 2017), which is beyond the capability of our model. This means that other extensions of calving parameterizations are needed. But this is not particularly relevant to the MICI mechanism in our opinion. We add this text:

Winter calving can occur in later winter (Cassotto et al., 2015) when calving front height is at its annual minimum and presumably at its least vulnerable to structural failure. Hence, MICI (Marine Ice Cliff Instability) cannot explain this type of calving, and winter calving is specifically excluded from parametrizations of MICI (Pollard et al., 2018). The existence of winter calving has greatly reduced the range of seasonal fluctuations in front position, which inhibited the growing of a temporary ice shelf that would buttress the grounded ice. Thus, lack of winter calving would cause underestimation of dynamic thinning as the glacier grows in winter.

Our calving criterion does not predict lower calving rates as the glacier retreats into its over-deepened basin. Although calving front height keep increasing during retreats, dynamic thinning rises highly non-linearly along with it, which leads to formation of a thin shelf which is then vulnerable to calving.

Our modeled retreats are not in contradiction to MISI (Marine Ice Shelf Instability). Most of the glacier buttressing is from the lateral margins and not the bed, this means that e.g. the advance of the glacier in 2018 due to ocean cooling does not contradict the MISI either since the calving position is not only governed by bed geometry. The cessation of retreat during the later part of our ~ 60 year simulation can be attributed to the strong flow convergence near the glacier front that largely offsets the dynamic thinning (Fig. 9). Notice that the south side of the fast-flow-area in last century is quite close to ice-free land while in later half of this century convergent flow in the south is fed by a substantial area of ice stream. We add this point to the text. We discuss all these features of calving in section 4.

[Figure]

Figure 9. Simulated velocity vectors in 2004 (pink vectors) with their magnitudes (right color bar)

**and velocity difference between 2004 and 2099 (2099's minus 2004's, black vectors), for clarity vector lengths are clipped at 5 km a⁻¹.**

*The role of melange and its parameterization are said to have a relatively large impact on the results, but the exact role of melange and the associated processes that could impact calving remains unclear. What happens to the simulations when the buttressing provided by the ice melange is removed?*

**Reply:** Ice mélange buttressing effects played a decisive role in the recent retreating and evolving of Jakobshavn (Joughin et al., 2004; Joughin et al., 2008; Vieli et al., 2011; Nick et al., 2013). In our model, mélange buttressing does affect calving by altering the stress field that contributes to crevasse penetration depth (Eq. 12), with the sensitivity of the corresponded parameter α tested (Section 2.4). This paper aims at reproducing the evolutions of Jakobshavn in the real world where mélange buttressing matters, for example we add: Our buttressing parameterization gives a longitudinal resistance that is 18% of the driving force at calving front (Eq. 10), for the instance of 2004.

*The processes included in the simulations include many parameters, and these parameters are not always justified or explained. In particular what they physically represent and what the impact is for the simulations. For example: How much buttressing does the melange represent? What is the equivalent ice thickness needed to get a similar buttressing? What is the tuning scalar for the run-off in the crevasses? What ocean temperatures are used for the forcing, and how was this choice made? How are the ocean temperature converted from the far field, to the fjord and to the grounding line region?*

**Reply:** We explain our parameterization for external forcing into fine details in section 2.2. The verifications of our parameterizations are shown in Fig. 5 and discussed in section 4. We give additional details on various questions the referee raises:

Our buttressing parameterization gives a longitudinal resistance that is 18% of the driving force at calving front (Eq. 10), for the instance of 2004.

We tune parameters $\alpha$ (over the range 0.7−1.2 for $\alpha_1$ and 0.09-0.12 for $\alpha_2$) and $\beta$ (0.04 - 0.075) manually.

Local ocean circulation in Ilulissat fjord driven by buoyancy plume brings deep water from outside to the grounding line of Jakobshavn, and renews the fjord within 90 days in summer (Gladish et al., 2015). Generally, Jakobshavn's fjord is ~ 800 m deep but with a sill of only ~ 200 m depth at its entrance. The deepest water outside the sill can flow over the sill and reach the grounding line of
Jakobshavn (Gladish et al., 2015). We use 300 m depth ocean temperatures collected from a CTD site close to the mouth of Ilulissat fjord (Fig. 1) as an approximation of ocean temperatures near the glacier grounding line. A comprehensive study focusing on ocean circulation within Ilulissat fjord validated this approximation (Gladish et al. 2015).

$T_f$ is the far field ocean forcing temperature, taken in Disko Bay (CTD in Fig. 1), relative to pressure melting temperature under the ice shelf. Thus $T$ and $T_f$ are related simply by ice depth and salinity.

*The initialization is rather confusing, with a target date of 2004 at the beginning of the simulations, but other datasets with different times are used for the inverse problem (2012) and the relaxed surface elevation (1998). This part should be clarified to better understand the rationale behind the initialization procedure.*

**Reply:** Agreed, the statement was imprecise. We have rewritten this in response to other criticism on the method:

1) We solved the inverse problem for basal conditions (Eq. 7) and stiffening factor using 2010 velocities (Joughin et al., 2010) and 2009 geometry (Gogineni et al., 2012), following Cornford et al. (2015). Our friction coefficient and stiffening factor fields are shown in Fig. 3. Fig. S1 shows the discrepancy between observed velocity field (Joughin et al., 2010) and the velocity derived from the inversion.

2) Starting from the inversion of step 1, we let the model glacier evolve freely without calving and with zero SMB and with sub-shelf melting ($\gamma$=0.0238) forced by repeating the observed 2004 ocean temperature for 11 years until its surface elevation profile reached a state shown in Fig. S2.

3) We carried out several 10-year simulations each with different $\beta$ values estimated. These simulations were forced by repeatedly applying the 2004 seasonal climate forcing so that the glacier approaches a steady state. From these, we selected the $\beta$ that provided a calving front position closest to that observed in 2004. The best $\beta$ here is 0.034. This is our best guess for the 2004 state. The annual minimum extent of Jakobshavn retreats ~ 2 km from 2004 to 2005 following the loss of melange butressing, but then stabilizes until 2007 (Joughin et al. 2010). Annual maximum extents are stable over the 2004-2007 period. Front velocities increase slowly from 2004-2007 (~5.9% $a^{-1}$ Joughin et al. 2010), and the model simulated velocities increase by about 3% $a^{-1}$. This period of relative stability also makes 2004 a good time from which to start transient simulations.

Basal friction coefficient values downstream of the 2010 grounding line were set equal to that in the nearest 2010 grounded location. This was necessary because steps 2 and 3 involved grounding line advance beyond the region for which basal friction coefficients had been inferred. The geometry after this spin up procedure, and the friction coefficient and stiffening factor distribution from the inversion in step 1 were used as the initial condition for model calibration.

[Figure]

**Figure S1. A) Velocity discrepancy (velocity from inversion - observed) and B) the observed velocity field (Joughin et al., 2010).**

[Figure]

**Figure S2. Profiles of surface elevation during the initialization procedure (section 2.3) step 3. Black solid line and black dashed line show the known profiles taken in the 1990s (Bamber et al., 2001) and 2010 (Gogineni et al., 2012) respectively. The profile with legend '1st yr' is the final state of section 2.3 step 2. The profile '7th yr' is the geometry rebuilt for 2004's Jakobshavn, which is the initial state for later simulations.**

*I am wondering how reliable the GCMs are to reproduce the temporal patterns of variability of the glacier: to my knowledge, GCMs do not get the right timing for the variability, so maybe some reanalysis data would perform better for that.*

**Reply:** In model calibration, our forcing data comes from observations and MAR regional surface mass and energy balance model (Alexander et al. 2016) driven by the ERA-Interim reanalysis (Dee et al., 2011). The ERA-Interim reanalysis is widely used on Greenland by e.g. MAR and RACMO communities and is free of artifacts. Hence we do not reply on ESM (GCM) data for the historical variation prior to 2014.

*Finally, there are not many figures showing the results, e.g., the spatial distribution of ice front position at the end of the simulations for the different cases or the mass loss for the different cases (just a few numbers in the table). It would be a good addition to the paper to add a few figures to get a better sense of how this glacier could change in the future, such as the spread of results, the spatial evolution of the ice front, or the evolution of mass loss and discharge with time.*

**Reply:** Agreed. We add Fig. 7c and Fig. 8 to show Jakobshavn's retreats on 2-D plane view under the forcing for best guess, lower and upper bounds of mass loss projection.

[Figure]

**Figure 7. Modeled profiles of (A) January velocity and (B) January surface elevation along the center-flow-line (purple dash line in panel C) of Jakobshavn Isbræ from 2004 to 2099 for the RCP4.5 scenario. Bedrock elevation is shown in black. Black dotted line is the surface elevation profile extracted from radar data measured around 2010 (Gogineni et al., 2012). Profiles are shown at intervals of 1 years. Profiles are color-coded in the legend and range from blue to green and red. (C) Modeled July front positions (color bar) over its bedrock (grayscale bar) at intervals of 2 years.**

[Figure]

**Figure 8.** Upper and lower estimates of July front positions within this century with colors indicating the date (color bar) for A) lower bound with scalings of (1,0.8) and the HadGEM-ES forcing B) upper bound of mass loss projection with (α, γ) parameter scalings of (1.2,1), and the 7-model ensemble climate forcing.

**3 Specific comments**

*p.2 l.26: "with" → "leading to a"*
**Reply:** Done.

*p.2 l.26: "5.6 mm sea-level-rise" → "5.6 mm of sea level rise"*
**Reply:** Done.

*p.2 l.28: Why is the model unable to reproduce the winter calving? Is that a limitation of the model parameterization, the representation of calving (maybe the crevasse-depth calving is just one mode of calving and does not cover all the cases), the initial conditions? And what do you think are the consequences of the lack of winter calving?*

**Reply:** In our calving criterion, crevasse penetration depth depends on surface water run-off which equals to zero in winter, thus no calving occurs in simulation. As discussed in section 4.4, these winter calving might be the consequences of non-hydrostatic processes related to the opening of basal crevasses. The existence of winter calving has greatly reduced the range of seasonal fluctuations in front position, which inhibited the growing of a temporary ice shelf that would buttress the grounded ice. Thus, lack of winter calving would cause underestimation of dynamic thinning as the glacier grows in winter. We added some more text on this.

Winter calving can occur in later winter (Cassotto et al., 2015) when calving front height is at its annual minimum and presumably at its least vulnerable to structural failure. Hence, MICI (Marine Ice Cliff Instability) cannot explain this type of calving, and winter calving is specifically excluded from parametrizations of MICI (Pollard et al., 2018). The existence of winter calving has greatly reduced the range of seasonal fluctuations in front position, which inhibited the growing of a temporary ice shelf that would buttress the grounded ice. Thus, lack of winter calving would cause underestimation of dynamic thinning as the glacier grows in winter.

*p.2 Fig.1 caption: as mentioned above, using Jakobsson et al. (2012) is a rather un-expected choice, so it would be good to justify it and quantify the difference in bedrock elevation between this map and the other more standard maps.*

**Reply:** You may notice that Jakobsson et al. (2012) provides ocean bathymetry data (sea floor elevation). Our glacier geometry data (Gogineni et al., 2012) is derived from the same institution's products as used in BedMachine V3 with data processed by the Center for Remote Sensing of Ice Sheets (CReSIS, Leuschenetal., 2010 updated 2016). For computing resource considerations, we chose the earlier product because it has 500-meter-resolution. The differences between these two datasets are shown in Fig. S3.

[Figure]

**Figure S3. A) Bed elevation from BedMachine v3 (Morlighem et al., 2017) minus those from (Gogineni, 2012) used in this paper.**

*p.2 l.30: the 17km/a speed is a seasonal speed happening over a few months in summer and not an annual velocity, it would be good to mention that.*
**Reply:** Yes, we modify the sentence to say:
Jakobshavn Isbræ (Fig. 1) is Greenland's largest and fastest outlet glacier, with transient speeds of up to 17 km a$^{-1}$ (Joughin et al., 2014).

*p.3 l.44: "possessed" → "had"*
**Reply:** Done.

*p.3 l.57: "far faster" → "much faster"*
**Reply:** Done.

*p.4 l.64: "must be zero at the grounding line as it begins to float": I would rather say that it is zero under the floating tongue.*
**Reply:** Agreed. We change it to say: … it is zero at the grounding line as it begins to float.

*p.4 l.74: the melange does not really belong to the ice shelf or the glacier ("its floating melange"). I am also surprised to see "desintegration" associated to "melange" because the melange changes a lot seasonally, which is rather common, so I don't understand why these changes would be qualified of desintegration.*
**Reply:** We modify the sentence to say:
Loss of buttressing from the weakening mélange or enhanced submarine melting could have triggered the dramatic changes seen in Jakobshavn Isbræ at the end of the 20th century.

*p.4 l.80: There is also a new study on Jakobshavn by Bondzio et al. (2018) using a2d plan view model and a different calving parameterization, so it would be good to include these results in the comparison.*
**Reply:** At the end of this paragraph we added:
Bondzio et al. (2018) applied a similar calving model that remove any ice where tensile stress exceeds a threshold, as simulated with a SSA (Shallow Shelf Approximation) model, regardless of ice thickness. To represent seasonal fluctuation of front positon, their stress threshold is a stepwise constant function in time with low values in summer. After calibration, their model can closely reproduce the observed behavior from 1985 to 2018 when forced only with ocean temperatures.

*p.5 l.92: "BISICLES continuum ice sheet dynamics model" → "BISICLES ice sheet model"*
**Reply:** Done.

*p.5 l.100: "in hydrostatic equilibrium": the floating part only is in equilibrium"*
**Reply:** In BISICLES, the whole ice is in hydrostatic equilibrium (Cornford et al., 2013).

*p.6 l.107: "an approximate stress balance equation": replace by the name of the approximation and a reference as they are many difference approximations of the stress balance equations*

**Reply:** We modify this line to say:

An approximate stress balance equation (Schoof and Hindmarsh, 2010).

*p.8 l.136-137: How do you use the ocean conditions outside of the fjord to constrain the conditions inside the fjord and close to the grounding line?*

**Reply:** We add this sentence to the text:

Local ocean circulation in Ilulissat fjord driven by buoyancy plume brings deep water from outside to the grounding line of Jakobshavn, and renews the fjord within 90 days in summer (Gladish et al., 2015).

*p.8 l.140: "working hypothesis": there is not much in the discussion addressing this hypothesis and whether it was a valid one.*

**Reply:** We found good correlation between ocean temperatures on the ice-ocean interface and velocities of ice front (Fig 2). We further discuss its validation in section 4.1.

*p.8 l.148: calving has been shown to be the main driver of the velocity Bondzio et al. (2017), so that changes in calving front positions could explain most of the dynamic changes of Jakobshavn Isbrae over the past three decades, so that changes in basal conditions indeed have a small impact for this glacier.*

**Reply:** Agreed. Our basal drag coefficient is fixed in time.

*p.8 l.149-150: This is an interesting way to change the buttressing at the front. How different would the results be with another method to account for this buttressing?*

**Reply:** Our parameterization of mélange buttressing is similar to Nick et al. (2013), which also alters the stress balance at calving front. We have not run other simulation types, but the results could be compared with other authors, as we do in the discussion section.

*p.8 l.152: How much buttressing does this represent? What would be the equivalent ice thickness needed to get a similar buttressing?*

**Reply:** For example, in 2004 our buttressing parameterization gives a longitudinal resistance that equals to 0.18 times of the driving force at calving front (Eq. 10).

*p.9 l.156-157: What is this runoff symbol?*

**Reply:** We changed this symbol to '$R$'.

*p.9 l.163 and l.175: How do you link the far ocean field temperatures to the ocean temperature in the fjord and then the temperature at the grounding line?*

**Reply:** We add this sentence to the text:

$T_f$ is the far field ocean forcing temperature, taken in Disko Bay (CTD in Fig. 1), relative to pressure melting temperature under the ice shelf. Thus $T$ and $T_f$ are related simply by ice depth and salinity

*p.10 l.178: The depth of the ocean temperatures used should depend on the geometry of the fjord, including the highest depths of the sills. What is the depth of the ocean floor in Jakobshavn's fjord, and are there sills blocking the entry of the deepest waters?*
**Reply:** Yes. We add:
Local ocean circulation in Ilulissat fjord driven by buoyancy plume brings deep water from outside to the grounding line of Jakobshavn, and renews the fjord within 90 days in summer (Gladish et al., 2015). Generally, Jakobshavn's fjord is ~ 800 m deep but with a sill of only ~ 200 m depth at its entrance. The deepest water outside the sill can flow over the sill and reach the grounding line of Jakobshavn (Gladish et al., 2015).

*p.10 l.186: "last 2 decades" while the rest of the paper rather show results since 2004.*
**Reply:** "last 2 decades" is now "last decade".

*p.10 l.187: "bedrock topography and ice thickness data in the year 2009 come from Gogineni et al. (2012)": As mentioned above, why use this dataset and not the more recent Bamber et al. (2013) or Morlighem et al. (2017) topography? Also, is this the same dataset as Jakobsson et al. (2012) shown of figure 1?*
**Reply:** Our glacier geometry data (Gogineni et al., 2012) is derived from the same institution's products as used in BedMachine V3 with data processed by the Center for Remote Sensing of Ice Sheets (CReSIS, Leuschenetal., 2010 updated 2016). For computing resource considerations, we chose the earlier product because it has 500-meter-resolution. You may notice that Jakobsson et al. (2012) provides ocean bathymetry data (sea floor elevation).

*p.10 l.192-193: I am not sure to agree with this statement: the glacier was continuing to change following its ice tongue collapse as shown in Joughin et al. (2012).*
Lines are: "The aim of this initialization was provide a state rather similar to 2004, that is barely retreating on inter-annual scales (Joughin et al., 2010) and small changes of annual mean velocity in the following 3 years. Therefore"
**Reply:** Agreed, the statement was imprecise. The annual minimum extent of Jakobshavn retreats ~ 2 km from 2004 to 2005 following the loss of melange butressing, but then stabilizes until 2007 (Joughin et al. 2010). Annual maximum extents are stable over the 2004-2007 period. Front velocities increase slowly from 2004-2007 (~5.9% a[-1] Joughin et al. 2010), and the model simulated velocities increase by about 3% a[-1]. This period of relative stability also makes 2004 a good time from which to start transient simulations.

*Fig.3: the stiffening factor inferred with inverse problems is often difficult to physically explain and it is here mostly equal to 1. How different would the results be if it was just assumed to be equal to 1 everywhere and the basal traction coefficient was adjusted accordingly?*

**Reply:** The stiffening factor deviates significantly from 1 in the fast flow region (Fig 3). We are satisfied with our inversion results, which is verified in the new Fig. S1. Our inversion followed the standard built-in procedures in BISICLES (Conford et al., 2015). Here we do not want to explore techniques of solving the inverse problem, the paper is already quite long and reasonably complex, and we think it stands on its own without this.

[Figure]

Figure S1. A) Velocity discrepancy (velocity from inversion - observed) and B) the observed velocity field (Joughin et al., 2010).

*p.11 l.199-200: Why are the velocity from 2010 and the geometry from 2009 used in the inverse problem why the simulation is initialized to reproduce 2004?*
**Reply:** There are no thickness data available for 2004. Our initializing procedure aims at rebuilding the geometry of 2004 from the closest dataset available (Gogineni et al., 2012).

*p.12 l.201: "friction coefficient" and stiffening factor*
**Reply:** Agreed. Done.

*p.12 l.210: Why is the model run until the profile matched the 1998 profile? I thought the target date was 2004?*
*p.12 l.211: How is it changed exactly?*
*p.12 l.214: Are you trying to get a stable state (How do you define stable by the way? What are the variables considered?) or to match the 2004 front position? I am also a little surprised that you are mentioning a "stable state" as the glacier has been continuously since at least the 90's.*
**Reply:** These questions are moot now as we rewrote section 2.3 (see above) in response to both yours and the other referee's confusion.

*p.14 l.254: Why is considered to be the total calving? Is it the difference between the ice front positions in 2013 and 2004 or the sum of the annual ice front change position?*
**Reply:** Line 254 now becomes:

1. Total calving front retreat from 2004-2013 measured by the difference between 2004 and 2013's annual maximum extent.

*p.16 Fig.5a: There was an earlier mention of the relative stability after 2004, but this figure actually tends to show that there is not much stability at this period, with the summer front position retreating more every year.*

**Reply:** Agreed. We have rewritten this: The annual minimum extent of Jakobshavn retreats ~ 2 km from 2004 to 2005 following the loss of melange butressing, but then stabilizes until 2007 (Joughin et al. 2010). Annual maximum extents are stable over the 2004-2007 period. Front velocities increase slowly from 2004-2007 (~5.9% a$^{-1}$ Joughin et al. 2010), and the model simulated velocities increase by about 3% a$^{-1}$. This period of relative stability also makes 2004 a good time from which to start transient simulations. December front positions during 2004-2006 are close to stable, which are reproduced in our initialization procedure step 3. Notice observation extends no further prior to the summer of 2004.

*p.18 Fig.6: Is it the bottom or the 300 m depth temperature?*
**Reply:** The 300 m depth. We see the confusion with the right hand y-axis mis-labelled in Fig. 6. This has been corrected.

*p.19 l.312, 314, 318: What GCM was used to force RACMO?*
**Reply:** We change the sentence to include HadGEM2-ES here.
**Figure 6. Climate forcing for future projection under the RCP4.5 scenario taken as 300 m depth ocean temperatures from HadGEM2-ES (orange) compared with the ensemble mean (red) of 7 Earth System Models (HadGEM2-ES, BNU-ESM, MIROC-ESM, IPSL-CM5A-LR, CSIRO-Mk3L-1-2, NorESM1-M and MPI-ESM-LR), (right axis), with their linear trends. Annual maximum monthly surface water run-off near Jakobshavn Isbrae's terminus from RACMO (forced by outputs from HadGEM2-ES) is shown in blue.**

*p.19 l.303: change reference: Joughin et al. (2010) probably did not guess what would happen in the 2013-2017 period.*
**Reply:** Yes, but the dataset is updated to include the more recent period and is still cited as Joughin et al. (2010). https://nsidc.org/data/nsidc-0481. So we think our usage is quite standard.

*p.20 l.326-329: are these results shown on a figure?*
**Reply:** Yes. Please see new Fig. 7.

[Figure]

Figure 7. Modeled profiles of (A) January velocity and (B) January surface elevation along the center-flow-line (purple dash line in panel C) of Jakobshavn Isbræ from 2004 to 2099 for the RCP4.5 scenario. Bedrock elevation is shown in black. Black dotted line is the surface elevation profile extracted from radar data measured around 2010 (Gogineni et al., 2012). Profiles are shown at intervals of 1 years. Profiles are color-coded in the legend and range from blue to green and red. (C) Modeled July front positions (color bar) over its bedrock (grayscale bar) at intervals of 2 years.

*p.20 Table 1 and lines 335-341: the numbers in the table and in the paragraph are somehow different (2068 vs 2029 Gt of mass loss by 2100 for example), but I might have missed something. It would also be appropriate to add the results from Bondzio et al. (2018) in the comparison here.*

**Reply:** Sorry, 2029 should be 2068. We add a sentence in this paragraph:

Another SSA model (Bondzio et al., 2018) projects larger retreats than ours, and uses a calving parameterization that predicts the location of calving depending on tensile stress distribution, regardless of ice thickness. Their calving criterion implies a nonlinear relationship between crevasse depth and stress, compared with our linear dependence (Eq. 12). We suggest that this nonlinearity might lead to overestimated retreats.

*p.22 l.384: Does the fast flow go all the way to the sides of the fjord? It is a bit surprising to me that the bedrock provides so little resistance compared to the sides.*
**Reply:** The basal trough beneath Jakobshavn is quite deep and narrow, which makes large velocity gradients perpendicular to flow direction at the two sides of the trough.

*p.22 l.399: Is it possible to test this assumption of the impact of the shear-margin weakening mechanism?*
**Reply:** Not possible because we cannot measure the ice viscosity directly. We made cross model comparison between ours and Bondzio et al., (2016).

*p.26 l.448: There may also be some limitations associated with the initial conditions (ice too thick close to the ice front as shown on Fig.7).*
**Reply:** Yes. We add a sentence here:
One reason for the discrepancy may be errors in initial ice thickness and real geometry in 2004.

*p.26 l.462: "stimulate" → "simulate"*
**Reply:** Done.

*p.28 l.496: Did you use each GCM individually or used the mean of the 7 models?*
**Reply:** We modify this sentence to say:

[revised manuscript text omitted]

---

## Referee Report (RR1)

**Review of "Simulated retreat of Jakobshavn Isbæduring the 21$^{st}$ century"**

**1 Summary statememement**

The new version of the manuscript "Simulated retreat of Jakobshavn Isbæduring the 21$^{st}$ century" by X. Guo and colleagues adresses most of the comments raised by the three reviewers, presents a more nuanced view of the model's ability to reproduce the evolution of this outlet glacier as well as its limitations, and provides more information to better put these new results in the context with previous studies.

The text is sometimes confusing, especially in the abstract and does not clearly state the results in terms of what are the new parts revealed by the study, and what confirms previous results obtained y earlier studies. This should be better emphasized in the text.

**2 Major comments**

Overall, it is not always clear when this manuscript agrees with previous studies, and when it disagrees or new elements are demonstrated by this new modeling study. It would be good to more clearly distinguish and compare with previous papers.

There is a hypothesis made on p.9 (l.156) about the correlation between the sea ice and the flow speed near the terminus. There is no clear assessment of or conclusion about this hypothesis later on the paper. It would be useful to validate it (or not) in light of the modeling results obtained in the manuscript.

Is the initial state chosen of the model (representing a combination of datasets taken at different time but close to 2009 followed by a relaxation with zero SMB but some sub-shelf melt) close to the 2004 conditions (p.12 l.225)?

There is no crevasse depth in the Von Mises tensile criterion (p.29 l.487), so the comparison made between the results obtained with a crevasse depth criterion and the Von Mises tensile stress criterion is not accurate, and needs to be improved.

It is explained that the winter calving happens late in the winter season, "when calving front height is at its annual minimum and presumably at its least vulnerable to structural failure". I don't understand why calving would happen during this period if ice close to the front is "least vulnerable" during this period.

**3   Technical comments**

p.1 l.13: rephrase, not clear

p.1 l.20: rephrase, not clear

p.3 l.44: Are you talking about the "ice mélange" in front the "ice front" or about the "ice tongue" in front of the "grounding line". The ice mélange does not below to the glacier and its length varies a lot seasonally, so I don't understand how Jakobshavn "had a $\sim 15$ km long floating ice mélange. Please rephrase.

p.3 l.47: [?] esimated the basal melt rate, not the thinning rate.

p.5 l.88: the ocean does melt the ice, which is only one process by which the ice can thin.

p.5 l.93: "method" $\rightarrow$ "results"

p.6 l.106: only the ice tongue is in hydrostatic equilibrium

p.8 l.148: "fjord" $\rightarrow$ "fjord waters"

p.12 l.217: "was" $\rightarrow$ "is to"

p.12 l.236: "velocities" $\rightarrow$ "velocity"

p.14 l.262: "multiplying by" $\rightarrow$ "multiplying it by"

p.20 l.340: What does "higher" mean? Please rephrase.

p.23 l.372: "produces" $\rightarrow$ "produce"

p.29 l.487: there is no crevasse depth in the von mises tensile stress criterion.

p.30 l.510: "further extend a further" $\rightarrow$ "further extend"

p.31 p.531: The MICI depends on the ice thickness, and I don't think there is any consideration of season in its parameterization.

p.32 l.565: rephrase, not clear.

p.32 l.567: "Retreat slows" $\rightarrow$ "Retreat will slow"